# Diving into Kronecker Adapters: Component Design Matters

**Jiayu Bai**[1]  **Danchen Yu**[1]  **Zhenyu Liao**[1]  **TianQi Hou**[2]  **Feng Zhou**[3]  **Robert C. Qiu**[1]  **Zenan Ling**[1]

## Abstract

Kronecker adapters have emerged as a promising approach for fine-tuning large-scale models, enabling high-rank updates through tunable component structures. However, existing work largely treats the component structure as a fixed or heuristic design choice, leaving the dimensions and number of Kronecker components underexplored. In this paper, we identify component structure as a key factor governing the capacity of Kronecker adapters. We perform a fine-grained analysis of both the dimensions and number of Kronecker components. In particular, we show that the alignment between Kronecker adapters and full fine-tuning depends on component configurations. Guided by these insights, we propose Component Designed Kronecker Adapters (CDKA). We further provide parameter-budget–aware configuration guidelines and a tailored training stabilization strategy for practical deployment. Experiments across various architectures and modalities demonstrate the effectiveness of CDKA. Code is available at https://github.com/rainstonee/CDKA.

## 1. Introduction

Parameter-Efficient Fine-Tuning (PEFT) methods (Houlsby et al., 2019; Li & Liang, 2021; Liu et al., 2022; Fu et al., 2023; He et al., 2023; Hu et al., 2022) have achieved state-of-the-art performance in adapting large-scale pretrained models (Devlin et al., 2019; Raffel et al., 2020; Brown et al., 2020; Achiam et al., 2023; Touvron et al., 2023; Rombach et al., 2022; Kirillov et al., 2023) to downstream tasks. As the most widely adopted approach in PEFT, adapter-based methods (Houlsby et al., 2019; Pfeiffer et al., 2021; He

et al., 2022; Hu et al., 2022; Meng et al., 2024; Zhang et al., 2025) incorporate existing network layers with lightweight adapters containing only a small number of trainable parameters. Despite their efficiency, adapter-based methods typically exhibit a noticeable performance gap compared to full fine-tuning on complex tasks (Hu et al., 2022; Ding et al., 2023; Liu et al., 2024; Biderman et al., 2024; Wang et al., 2025; Zhang et al., 2025). This gap largely arises because adapters are constrained to limited expressive spaces, such as low-rank subspaces in LoRA (Hu et al., 2022).

To mitigate this limitation, a growing body of work (Hyeon-Woo et al., 2022; Edalati et al., 2022; Ren et al., 2024; Li et al., 2025; Huang et al., 2025) has explored alternative adapter formulations beyond the simple matrix product used in LoRA, enabling more expressive and flexible representations. One notable direction among these extensions is the incorporation of the Kronecker product (Edalati et al., 2022; Braga et al., 2024; YEH et al., 2024; Yu et al., 2025; Sadeghi et al., 2025), which enables high-rank weight updates with minimal parameter budget. In this work, we consider a general formulation of Kronecker adapters, where the update on the pre-trained weight $\boldsymbol{W}_0$ is expressed as a sum of $r$ Kronecker components, namely:

$$\boldsymbol{W} = \boldsymbol{W}_0 + \sum_{i=1}^{r} \boldsymbol{B}^{(i)} \otimes \boldsymbol{A}^{(i)},$$

where $\boldsymbol{A}^{(i)} \in \mathbb{R}^{r_1 \times \frac{d_{\text{in}}}{r_2}}$ and $\boldsymbol{B}^{(i)} \in \mathbb{R}^{\frac{d_{\text{out}}}{r_1} \times r_2}$. Previous studies (Edalati et al., 2022; Sadeghi et al., 2025) have shown that the dimensions of the Kronecker components $\boldsymbol{A}^{(i)}$ and $\boldsymbol{B}^{(i)}$, which are governed by hyperparameters $r_1$ and $r_2$, together with the number of components $r$, play a crucial role in determining the expressive capacity of Kronecker adapters. We refer to the choice of $r_1$, $r_2$ and $r$ as *component design* for Kronecker adapters. Despite recent progress, substantial gaps remain in understanding how component design determines both the theoretical properties and empirical performance of Kronecker adapters. In practice, most existing approaches (Yu et al., 2025; Sadeghi et al., 2025) adopt component configurations that enable Kronecker adapters to approximate full-rank updates. However, their empirical performance still falls significantly short of full fine-tuning and is even inferior to LoRA, which is explicitly constrained to a low-rank subspace.

[1]School of Electronic Information and Communications, Huazhong University of Science and Technology [2]Huawei [3]Center for Applied Statistics and School of Statistics, Renmin University of China. Correspondence to: Zenan Ling <lingzenan@hust.edu.cn>.

*Proceedings of the 43rd International Conference on Machine Learning*, Seoul, South Korea. PMLR 306, 2026. Copyright 2026 by the author(s).

Our fundamental objective is to fully unlock the potential of Kronecker adapters through principled component design. Specifically, we seek to address:

- *whether component design is the key factor for Kronecker adapters* and

- *whether a principle exists for component design.*

In this paper, we provide a positive answer to these questions. We begin by highlighting the central role of component design in Kronecker adapters. We emphasize that the performance of Kronecker adapters does not consistently improve as the attainable rank increases. Instead, it exhibits distinct trends as $r_1$, $r_2$, and $r$ vary. To understand how individual component configurations influence the behavior of Kronecker adapters, we conduct a theoretical analysis grounded in the Kronecker singular value decomposition. We show that the subspace alignment between Kronecker adapters and full fine-tuning is fully determined by the choice of $r_1$, $r_2$, and $r$. Guided by these theoretical insights, we derive principles for component design and empirically validate their effectiveness. We refer to our approach as Component Designed Kronecker Adapters (CDKA). To facilitate the practical deployment of CDKA, we provide guidelines for selecting $r_1$, $r_2$, and $r$ under a fixed parameter budget. Furthermore, we propose a stabilization strategy tailored to CDKA, which further enhances its performance.

To validate the effectiveness of CDKA, we conduct experiments across various Natural Language Processing (NLP) and Computer Vision (CV) tasks, including Natural Language Understanding (NLU), mathematical reasoning, code generation and image classification. Notably, CDKA achieves state-of-the-art performance on mathematical reasoning and image classification, while attaining the second best performance on code generation. On NLU tasks, CDKA attains near-optimal results using only $12.5\%$ of the trainable parameters. More importantly, CDKA substantially improves the performance of Kronecker adapters, making them competitive with the strongest PEFT approaches.

Our contributions are summarized as follows.

- We emphasize that the performance of Kronecker adapters depends critically on the choice of component dimensions, which are determined by $r_1$ and $r_2$, and the number of components $r$. Through a theoretical analysis based on Kronecker singular value decomposition, we show that the subspace alignment with full fine-tuning is governed by these choices. Based on this analysis, we derive principles for component design and empirically validate their effectiveness.

- Guided by these theoretical insights, we propose Component Designed Kronecker Adapters (CDKA). We

provide guidelines for component design under a fixed parameter budget. We further introduce a training stabilization strategy tailored to CDKA, leading to consistent and improved empirical performance.

- We validate CDKA across a range of NLP and CV tasks, showing that it achieves state-of-the-art performance on mathematical reasoning and image classification, the second best results on code generation, and near-optimal performance on NLU using only $12.5\%$ of the trainable parameters. More importantly, CDKA substantially improves the performance of Kronecker adapters, rendering them competitive with state-of-the-art PEFT methods.

### 1.1. Related Works

**Adapter-based methods.** As one of the most widely used and effective adapter-based methods, Low-Rank Adaptation (LoRA) (Hu et al., 2022) assumes that changes in the weights of pretrained models exhibit a low-rank structure. Accordingly, LoRA approximates weight updates by decomposing them into the product of two low-rank matrices. Numerous variants have been proposed to further improve the performance of LoRA. AdaLoRA (Zhang et al., 2023) adaptively allocates ranks, assigning higher capacity to more important components. rsLoRA (Kalajdzievski, 2023) introduces a carefully designed scaling factor to ensure training stability. LoRA-GA (Wang et al., 2024) approximates the gradients of full fine-tuning through initialization. LoRA-One (Zhang et al., 2025) proposes an improved initialization strategy inspired by theoretical analysis on the subspace alignment with full fine-tuning.

Beyond the standard matrix product formulation of vanilla LoRA, several approaches adopt more expressive adaptation mechanisms. DoRA (Liu et al., 2024) decomposes pretrained weights into magnitude and direction and applies LoRA to directional updates to enhance representational capacity. MELoRA (Ren et al., 2024) trains a collection of lightweight LoRA modules, each with a small parameter budget. FouRA (Borse et al., 2024) applies LoRA in the frequency domain, while HiRA (Huang et al., 2025) connects updated and pretrained weights via Hadamard product.

**Kronecker adapters.** KronA (Edalati et al., 2022) generalizes LoRA by replacing the standard matrix product with a single Kronecker product, thereby significantly reducing the number of parameters while enabling higher effective rank. LoKr (YEH et al., 2024) further extends this idea by incorporating an additional low-rank decomposition on the Kronecker component to improve expressive capacity. MoKA-MoE[1] (Yu et al., 2025) models the Kronecker com-

---

[1]We refer to the MoKA in Yu et al. (2025) as MoKA-MoE to distinguish it from the MoKA in Sadeghi et al. (2025).

ponent using Mixture-of-Experts, whereas MoKA (Sadeghi et al., 2025) represents the weight update as a mixture of Kronecker products with different component dimensions. Both approaches enhance the expressive capacity of the original KronA formulation.

## 1.2. Notations

For a matrix $\boldsymbol{K}$, we denote its spectral norm and Frobenius norm by $\|\boldsymbol{K}\|_2$ and $\|\boldsymbol{K}\|_F$, respectively. We use $\boldsymbol{U}_r(\boldsymbol{K})$ to denote the top-$r$ left singular subspace of $\boldsymbol{K}$, and $\boldsymbol{U}_{r,\perp}(\boldsymbol{K})$ to denote its orthogonal complement. Similarly, $\boldsymbol{V}_r(\boldsymbol{K})$ and $\boldsymbol{V}_{r,\perp}(\boldsymbol{K})$ denote the top-$r$ right singular subspace of $\boldsymbol{K}$ and its orthogonal complement, respectively. We define the vectorization operator by $\mathrm{vec}(\cdot)$. We denote the input and output dimensions of the adapters by $d_{\mathrm{in}}$ and $d_{\mathrm{out}}$, respectively. Throughout the paper, we assume by default that the component configurations satisfy $(r_1 \bmod d_{\mathrm{out}}) = (r_2 \bmod d_{\mathrm{in}}) = 0$.

## 2. Preliminaries

### 2.1. Low-Rank Adapters

Low-Rank Adaptation (LoRA) (Hu et al., 2022) is a state-of-the-art adapter-based method designed for linear layers in large-scale models. Rather than updating the full weight matrix $\boldsymbol{W} \in \mathbb{R}^{d_{\mathrm{out}} \times d_{\mathrm{in}}}$, LoRA introduces two low-rank matrices, $\boldsymbol{A} \in \mathbb{R}^{r \times d_{\mathrm{in}}}$ and $\boldsymbol{B} \in \mathbb{R}^{d_{\mathrm{out}} \times r}$, such that the weight update is expressed as

$$\boldsymbol{W} = \boldsymbol{W}_0 + \Delta\boldsymbol{W} = \boldsymbol{W}_0 + \boldsymbol{B}\boldsymbol{A}, \qquad (1)$$

where $\boldsymbol{W}_0$ denotes the pre-trained weight, which remains frozen during training. This formulation enables efficient adaptation to downstream tasks while requiring substantially fewer trainable parameters.

### 2.2. Kronecker Adapters

KronA (Edalati et al., 2022) first introduces the Kronecker product into the adapter-based framework. Unlike vanilla LoRA, which relies on a standard matrix product, KronA employs a single Kronecker product to enable higher rank updates while significantly reducing the number of trainable parameters. In this paper, we consider a more general formulation of Kronecker adapters, namely:

$$\boldsymbol{W} = \boldsymbol{W}_0 + \Delta\boldsymbol{W} = \boldsymbol{W}_0 + \sum_{i=1}^{r} \boldsymbol{B}^{(i)} \otimes \boldsymbol{A}^{(i)}. \qquad (2)$$

Here, $\boldsymbol{A}^{(i)} \in \mathbb{R}^{r_1 \times \frac{d_{\mathrm{in}}}{r_2}}$ and $\boldsymbol{B}^{(i)} \in \mathbb{R}^{\frac{d_{\mathrm{out}}}{r_1} \times r_2}$ for $\forall i \in [1, r]$, where $r_1$ and $r_2$ are hyperparameters that determine the dimensions of the Kronecker components. We refer to the selection of $r_1$, $r_2$ and $r$ as the *component design* for Kronecker adapters. As summarized in Table 1, both LoRA

and KronA are special cases of Eq. (2) with additional constraints. This formulation is driven by the Kronecker product singular value decomposition (Van Loan, 2000; Batselier & Wong, 2017), as formalized in Definition 2.1.

**Definition 2.1.** For a matrix $\boldsymbol{K} \in \mathbb{R}^{d_{\mathrm{out}} \times d_{\mathrm{in}}}$, its Kronecker product singular value decomposition is given by:

$$\boldsymbol{K} = \sum_{i=1}^{r^*} \sigma_i \boldsymbol{B}^{(i)} \otimes \boldsymbol{A}^{(i)}, \qquad (3)$$

where $\boldsymbol{A}^{(i)} \in \mathbb{R}^{r_1 \times \frac{d_{\mathrm{in}}}{r_2}}$ and $\boldsymbol{B}^{(i)} \in \mathbb{R}^{\frac{d_{\mathrm{out}}}{r_1} \times r_2}$, if and only if

$$\widetilde{\boldsymbol{K}} = \mathrm{Kreshape}(\boldsymbol{K}) = \widetilde{\boldsymbol{A}}\boldsymbol{\Sigma}\widetilde{\boldsymbol{B}}^{\top} \qquad (4)$$

is the singular value decomposition of $\widetilde{\boldsymbol{K}}$, where $\widetilde{\boldsymbol{A}} = [\mathrm{vec}(\boldsymbol{A}^{(1)}), \cdots, \mathrm{vec}(\boldsymbol{A}^{(r^*)})]$, $\widetilde{\boldsymbol{B}} = [\mathrm{vec}(\boldsymbol{B}^{(1)}), \cdots, \mathrm{vec}(\boldsymbol{B}^{(r^*)})]$ and $\boldsymbol{\Sigma} = \mathrm{diag}(\sigma_1, \cdots, \sigma_{r^*})$. The function $\mathrm{Kreshape}(\cdot)$ is defined in Definition E.1.

Despite recent progress, component design for Kronecker adapters remains challenging. KronA (Edalati et al., 2022) enforces $r_1 = r_2$ and $r = 1$ in practice to minimize the parameter budget, but this overly restrictive setting leads to inferior performance. MoKA-MoE (Yu et al., 2025) adopts the same configuration as KronA and incorporates a Mixture-of-Experts mechanism into the Kronecker components, which improves performance at the cost of substantial parameter budget and computational overhead. The work most closely related to ours is MoKA (Sadeghi et al., 2025), which models Kronecker adapters as a sum of Kronecker components with heterogeneous choices of $r_1$ and $r_2$ across different components. However, MoKA does not provide guidelines for such design and instead relies on manual adjustment, making it difficult to deploy in practice. More importantly, these studies fail to elucidate how the choices of $r_1$, $r_2$ and $r$ influence the performance of Kronecker adapters.

## 3. Main Results

In Section 2.2, we formulate component design as the selection of three hyperparameters: $r_1$ and $r_2$, which control the dimensions of the Kronecker components, and $r$, which determines the number of components. In this section, we systematically demonstrate and examine the full potential of Kronecker adapters through component design. In Section 3.1, we emphasize the central role of component design in Kronecker adapters. In Section 3.2, we investigate how component design governs the theoretical alignment between Kronecker adapters and full fine-tuning, and derive corresponding principles for effective design. In Section 3.3, we provide guidelines for selecting component configurations under a fixed parameter budget and further introduce a tailored training stabilization strategy.

*Table 1.* Constraints on the component configurations in previous adapter-based methods.

| Method | Constraint on $r_1$ and $r_2$ | Constraint on $r$ |
|---|---|---|
| Full fine-tuning | $r_1, r_2 = 1, d_{\text{in}}$ or $r_1, r_2 = d_{\text{out}}, 1$ $r_1 = r_2 = 1$ | No constraint $r \geq \min(d_{\text{in}}, d_{\text{out}})$ |
| LoRA (Hu et al., 2022) | $r_1 = r_2 = 1$ | $r < \min(d_{\text{in}}, d_{\text{out}})$ |
| KronA (Edalati et al., 2022) | $r_1 = r_2$ | $r = 1$ |
| Ours | No constraint | No constraint |

### 3.1. Component Design Matters for Kronecker Adapters

**Kronecker adapters enable higher rank.** We first identify the fundamental advantage of Kronecker adapters, which can achieve higher rank updates through component design under a fixed parameter budget. Under the formulation in Eq. (2), the number of trainable parameters in $\Delta\boldsymbol{W}$ can be expressed by:

$$\text{param}(\Delta\boldsymbol{W}) \propto r\left(\frac{r_1}{r_2} + \frac{r_2}{r_1}\right). \quad (5)$$

We define the maximum attainable rank of $\Delta\boldsymbol{W}$ as the highest possible rank achievable by $\Delta\boldsymbol{W}$, denoted by $\overline{\text{rank}}(\Delta\boldsymbol{W})$. Since $\Delta\boldsymbol{W}$ is a sum of $r$ Kronecker components, its rank is upper bounded by the sum of the ranks of individual components, namely:

$$\begin{aligned}
\overline{\text{rank}}(\Delta\boldsymbol{W}) &= \sum_{i=1}^{r} \overline{\text{rank}}(\boldsymbol{B}^{(i)} \otimes \boldsymbol{A}^{(i)}) \\
&= r\,\overline{\text{rank}}(\boldsymbol{B}^{(i)})\overline{\text{rank}}(\boldsymbol{A}^{(i)}) \\
&= rr_1 r_2,
\end{aligned} \quad (6)$$

where we use the property $\text{rank}(\boldsymbol{B}^{(i)} \otimes \boldsymbol{A}^{(i)}) = \text{rank}(\boldsymbol{B}^{(i)})\text{rank}(\boldsymbol{A}^{(i)})$ and assume each component attains ranks $r_1$ and $r_2$, respectively. Building upon Eq. (5) and Eq. (6), we obtain the following property for Kronecker adapters, as formalized in Remark 3.1.

*Remark* 3.1. Under a fixed parameter budget, Kronecker adapters can always achieve a higher attainable rank than vanilla LoRA by setting $r_1 = r_2 > 1$.

**Unleash the high rank potential through component design.** Through the above analysis, we observe that the maximum attainable rank of Kronecker adapters grows linearly with $r_1$, $r_2$, and $r$. However, this increase does not translate into consistent empirical gains. As shown in Table 2, performance can even deteriorate when the adapter reaches full rank. We attribute this discrepancy to the fundamentally different roles played by $r_1$, $r_2$, and $r$ in shaping empirical performance. Specifically, under the same attainable rank 8,

*Table 2.* Inconsistency in component design.

| $\overline{\text{rank}}(\Delta\boldsymbol{W})$ | $r_1, r_2, r$ | GSM8k |
|---|---|---|
| 4 | $2, 2, 1$ | $49.93_{\pm 1.25}$ |
| 4096 | $64, 64, 1$ | $49.00_{\pm 0.41}$ |
| 8 | $4, 2, 1$ | $49.58_{\pm 0.34}$ |
| 8 | $2, 4, 1$ | $50.45_{\pm 0.54}$ |
| 8 | $2, 2, 2$ | $51.58_{\pm 0.18}$ |

increasing $r$ or $r_2$ leads to clear performance improvements, whereas increasing $r_1$ degrades performance.

These observations indicate that the expressive capacity of Kronecker adapters cannot be fully characterized by the attainable rank alone. Instead, how the rank is realized through different component configurations plays a crucial role in determining empirical performance. Thus, rank should be viewed as a necessary but insufficient indicator of the capacity of Kronecker adapters. This motivates a more fine-grained analysis of the roles of $r_1$, $r_2$, and $r$ beyond their contribution to rank, which we investigate in the following section.

### 3.2. Component Designed Kronecker Adapters

In this section, we theoretically analyze how different component configurations influence the performance of Kronecker adapters, thereby providing principled insights into effective component design.

**Problem settings.** We investigate whether Kronecker adapters exhibit a similar "subspace alignment" with the first step gradient of full fine-tuning as vanilla LoRA (Zhang et al., 2025). More importantly, we aim to understand how this alignment property is affected by component design. To simplify the analysis, we consider a linear setting following seminal LoRA-related theoretical studies (Hayou et al., 2024; Zhang & Pilanci, 2024; Zhang et al., 2025), in which the loss of Kronecker adapters is defined as:

$$\mathcal{L}_{\text{KA}} = \frac{1}{2N}\|\boldsymbol{Y} - (\boldsymbol{W}_0 + \sum_{i=1}^{r} \boldsymbol{B}^{(i)} \otimes \boldsymbol{A}^{(i)})\boldsymbol{X}\|_F^2, \quad (7)$$

where $\boldsymbol{X} = [\boldsymbol{x}_1, \cdots, \boldsymbol{x}_N] \in \mathbb{R}^{d_{\text{in}} \times N}$ consists of $N$ i.i.d. input samples drawn from an isotropic, zero-mean sub-Gaussian distribution, and $\boldsymbol{Y} = [\boldsymbol{x}_1, \cdots, \boldsymbol{x}_N] \in \mathbb{R}^{d_{\text{out}} \times N}$ denotes the corresponding ground-truth outputs. The loss in Eq. (7) can be minimized using the following gradient descent updates with learning rate $\eta$:

$$
\begin{aligned}
\boldsymbol{A}_{t+1}^{(i)} &= \boldsymbol{A}_t^{(i)} - \eta \nabla_{\boldsymbol{A}_t^{(i)}} \mathcal{L}_{\text{KA}}, \\
\boldsymbol{B}_{t+1}^{(i)} &= \boldsymbol{B}_t^{(i)} - \eta \nabla_{\boldsymbol{B}_t^{(i)}} \mathcal{L}_{\text{KA}},
\end{aligned} \tag{8}
$$

where $\boldsymbol{A}_t^{(i)}$ and $\boldsymbol{B}_t^{(i)}$ denote the values of $\boldsymbol{A}^{(i)}$ and $\boldsymbol{B}^{(i)}$ after $t$ steps of gradient descent, respectively. Accordingly, the loss of full fine-tuning is given by:

$$
\mathcal{L}_{\text{full}} = \frac{1}{2N} \|\boldsymbol{Y} - \boldsymbol{W}\boldsymbol{X}\|_F^2. \tag{9}
$$

As a result, the first step gradient of full fine-tuning is:

$$
\boldsymbol{G}_0 = \frac{1}{N}(\boldsymbol{Y} - \boldsymbol{W}_0\boldsymbol{X})\boldsymbol{X}^\top. \tag{10}
$$

Building on the concept of Kronecker product singular value decomposition in Definition 2.1, we quantify the alignment between $\widetilde{\boldsymbol{A}}_t$ and the gradient $\boldsymbol{G}_0$ following Zhang et al. (2025) with[2]:

$$
\|\boldsymbol{U}_{r^*,\perp}^\top(\widetilde{\boldsymbol{G}}_0)\boldsymbol{U}_{r^*}(\widetilde{\boldsymbol{A}}_t)\|_2 \tag{11}
$$

where $r^*$ is the rank of $\widetilde{\boldsymbol{G}}_0 = \text{Kreshape}(\boldsymbol{G}_0)$, $\widetilde{\boldsymbol{A}}_t = [\text{vec}(\boldsymbol{A}_t^{(1)}), \cdots, \text{vec}(\boldsymbol{A}_t^{(r)})]$, $\boldsymbol{U}_{r^*}(\widetilde{\boldsymbol{A}}_t)$ denotes the top-$r^*$ left singular subspace of $\widetilde{\boldsymbol{A}}_t$, and $\boldsymbol{U}_{r^*,\perp}(\widetilde{\boldsymbol{G}}_0)$ denotes the orthogonal complement of the top-$r^*$ left singular subspace of $\widetilde{\boldsymbol{G}}_0$.

Following the above settings, we build the alignment between the top-$r^*$ left Kronecker singular subspaces of $\boldsymbol{G}_0$ and $\widetilde{\boldsymbol{A}}_t$ in Theorem 3.2. The full version of Theorem 3.2 and the corresponding proof are referred to Appendix B.

**Theorem 3.2** (A simplified version of Theorem B.1 with $r^* \le r < 2r^*$.)**.** *Under the settings described in Section 3.2 and taking $r^* \le r < 2r^*$, we consider random Gaussian initialization for $\widetilde{\boldsymbol{A}}_0$ with $[\widetilde{\boldsymbol{A}}_0]_{ij} \sim \mathcal{N}(0, \alpha^2)$ and zero initialization for $\widetilde{\boldsymbol{B}}_0$ with $[\widetilde{\boldsymbol{B}}_0]_{ij} = 0$ and*

$$
\alpha \le \left( \frac{\theta\xi\sqrt{r_2}}{24r\sqrt{r_1 d_{in}}} \right)^{\frac{3\kappa}{2}} \sqrt{\frac{\sigma_1(\widetilde{\boldsymbol{G}}_0)r_2}{94.5\sqrt{r}r_1 d_{in}}}, \tag{12}
$$

*where $\kappa$ is the condition number of $\widetilde{\boldsymbol{G}}_0$. Then if we run gradient descent for $t^*$ steps on the Kronecker adapter with:*

$$
t^* \lesssim \frac{\ln\left( \frac{24r\sqrt{r_1 d_{in}}}{\theta\xi\sqrt{r2}} \right)}{\ln\left( 1 + \eta\sigma_{r^*}(\widetilde{\boldsymbol{G}}_0) \right)}, \tag{13}
$$

_____________

[2] We refer the alignment of $\widetilde{\boldsymbol{B}}_t$ to Appendix C as it leads to the same principles as $\widetilde{\boldsymbol{A}}_t$.

*we have the following alignment for $\forall\theta \in (0,1)$:*

$$
\|\boldsymbol{U}_{r^*,\perp}^\top(\widetilde{\boldsymbol{G}}_0)\boldsymbol{U}_{r^*}(\widetilde{\boldsymbol{A}}_{t^*})\|_2 \le \theta, \tag{14}
$$

*with probability at least $1 - C_1 \exp(-d_{in}\frac{r_1}{r_2}) - (C_2\xi)^{r-r^*+1} - C_3 \exp(-r) - C_4 \exp(-N)$ for some universal constants $C_1$, $C_2$, $C_3$, $C_4$.*

*Remark* 3.3 (Principles for component design.)**.** Building on Theorem 3.2, we show that the theoretical alignment between Kronecker adapters and full fine-tuning depends highly on component design, i.e., the choice of $r_1$, $r_2$ and $r$. Specifically, we desire the upper bound of $\alpha$ in Eq. (12) to be sufficiently large, so that the empirical variance used in practice lies within this range. Meanwhile, we aim for the upper bound of $t^*$ in Eq. (13) to be as small as possible, in order to accelerate training convergence. Both metrics favor a component design in which $r_2$ is chosen as large as possible, while $r$ and $r_1$ are kept as small as possible, subject to the constraint $r \ge r^*$. Motivated by the insights from Theorem 3.2, we therefore derive fundamental principles for component design, which are summarized as follows.

1. Increasing $r_1$ tends to degrade the performance of Kronecker adapters.

2. Increasing $r_2$ consistently improves the performance of Kronecker adapters.

3. Increasing $r$ does not lead to a sustained improvement in the performance of Kronecker adapters.

**Validation for Our Proposed Principles.** Guided by the principle in Remark 3.3, we show that Kronecker adapters can be effectively improved through component design. We refer to our approach as Component Designed Kronecker Adapters (CDKA). To validate the practical applicability of these principles, we investigate how the empirical behavior of CDKA varies as $r$, $r_1$ and $r_2$ are individually adjusted. Specifically, we fine-tune LLaMA-2-7B (Touvron et al., 2023) on a 100K subset of MetaMathQA (Yu et al., 2024), and assess generalization on GSM8k (Cobbe et al., 2021). See Appendix F for more results.

- **Validation for $r_1$.** We examine the performance of CDKA as $r_1$ varies while holding $r$ and $r_2$ fixed, with results shown in Table 3. We observe a general degradation in the performance of CDKA as $r_1$ increases, suggesting the use of a small $r_1$.

- **Validation for $r_2$.** We examine the performance of CDKA as $r_2$ varies while holding $r$ and $r_1$ fixed. As shown in Table 4, the performance increases consistently with $r_2$, indicating the use of a large $r_2$.

*Table 3.* CDKA with different $r_1$ for fixed $r_2$ and $r$.

| $r_1$ | GSM8k |
|---|---|
| 2 | $\mathbf{49.93}_{\pm 1.25}$ |
| 4 | $49.58_{\pm 0.34}$ |
| 8 | $48.80_{\pm 1.09}$ |
| 16 | $49.89_{\pm 0.27}$ |

*Table 4.* CDKA with different $r_2$ for fixed $r_1$ and $r$.

| $r_2$ | GSM8k |
|---|---|
| 2 | $49.93_{\pm 1.25}$ |
| 4 | $50.45_{\pm 0.54}$ |
| 8 | $53.17_{\pm 0.43}$ |
| 16 | $\mathbf{53.93}_{\pm 0.46}$ |

*Table 5.* CDKA with different $r$ for fixed $r_1$ and $r_2$.

| $r$ | GSM8k |
|---|---|
| 1 | $49.93_{\pm 1.25}$ |
| 4 | $\mathbf{54.56}_{\pm 1.62}$ |
| 16 | $54.18_{\pm 0.72}$ |
| 64 | $54.26_{\pm 0.69}$ |

*Table 6.* CDKA with different $r_1$ and $r_2$ under the same parameter budget.

| $r_1, r_2, r$ | GSM8k |
|---|---|
| 2,2,8 | $\mathbf{56.71}_{\pm 0.38}$ |
| 4,4,8 | $56.58_{\pm 0.42}$ |
| 8,8,8 | $56.38_{\pm 0.90}$ |
| 16,16,8 | $56.56_{\pm 0.64}$ |

*Table 7.* CDKA with different $r$ and $r_2$ under the same parameter budget.

| $r_1, r_2, r$ | GSM8k |
|---|---|
| 2, 2, 8 | $\mathbf{56.71}_{\pm 0.38}$ |
| 2, 16, 2 | $55.17_{\pm 1.04}$ |
| 2, 2, 32 | $56.15_{\pm 0.75}$ |
| 2, 16, 8 | $\mathbf{57.95}_{\pm 0.43}$ |

*Table 8.* CDKA with different initialization strategies.

| Init Method | GSM8k |
|---|---|
| $\boldsymbol{A}^{(i)} \sim \text{Ku}, \boldsymbol{B}^{(i)} = 0$ | $\mathbf{56.71}_{\pm 0.38}$ |
| $\boldsymbol{A}^{(i)} = 0, \boldsymbol{B}^{(i)} \sim \text{Ku}$ | $55.12_{\pm 1.12}$ |
| $\boldsymbol{A}^{(i)} \sim \text{Kn}, \boldsymbol{B}^{(i)} = 0$ | $52.59_{\pm 0.85}$ |
| $\boldsymbol{A}^{(i)} = 0, \boldsymbol{B}^{(i)} \sim \text{Kn}$ | $51.86_{\pm 0.53}$ |

- **Validation for $r$.** We now examine the performance of CDKA as $r$ varies with fixed $r_1$ and $r_2$, with results reported in Table 5. We observe that performance improves substantially as $r$ increases initially, but begins to fluctuate as $r$ continues to grow. This suggests that when $r$ is small, i.e., $r < r^*$, increasing $r$ yields clear performance gains. However, once $r$ exceeds $r^*$, further increasing $r$ leads to performance instability and can even degrade performance.

### 3.3. Component Design in Practice

In this section, we provide practical guidelines for CDKA to further improve its performance. We follow the same experimental settings as described in Section 3.2. See Appendix F for more results.

**Guidelines for component design under fixed parameter budget.** Building on the principles in Remark 3.3, we now give guidelines for component design when the parameter budget $\text{param}(\Delta \boldsymbol{W})$ in Eq. (5) is fixed.

*Guideline for choosing $r_1$.* According to the principle stated in Remark 3.3, increasing $r_1$ leads to performance degradation, whereas increasing $r_2$ consistently improves performance. Since the parameter budget depends on the ratio between $r_1$ and $r_2$, we investigate how the performance of CDKA is affected when they are increased simultaneously. As shown in Table 6, the performance of CDKA remains stable, which is consistent with our theoretical insights. In practice, we empirically find that keeping a small $r_1 \in [2, 4]$ yields slight improvements across different settings.

*Guideline for choosing $r$ and $r_2$.* According to the principle in Remark 3.3, when $r \geq r^*$, increasing $r_2$ consistently improves CDKA's performance, whereas increasing $r$ can even lead to degradation. This suggests that, for small pa-

rameter budgets, it is important to first increase $r$ to reach $r \geq r^*$. Once the parameter budget is larger, increasing $r_2$ becomes more effective than further increasing $r$. As shown in Table 7, when $r < 8$, increasing $r_2$ yields less improvement than increasing $r$, whereas for $r > 8$, increasing $r_2$ is clearly more beneficial. In practice, we empirically find that setting $r^* \in [2, 8]$ serves well as a boundary condition for choosing $r$ and $r_2$ across different settings.

$\lambda$ **ensures training stability.** In practice, we observe that the training stability of CDKA is highly sensitive to the choice of $r_1$, $r_2$, and $r$, as shown in Figure 1a. Under certain component configurations, this instability can even lead to gradient collapse. To mitigate this issue, we introduce a scaling factor $\lambda$ that explicitly depends on $r_1$, $r_2$ and $r$ to ensure the gradient norm is irrelevant of component design, thereby leading to stable and consistent training across all component configurations. The resulting scaled weight update is given by:

$$\Delta \boldsymbol{W} = \lambda_{r_1, r_2, r} \sum_{i=1}^{r} \boldsymbol{B}^{(i)} \otimes \boldsymbol{A}^{(i)}. \qquad (15)$$

Building on this concept, we establish stability conditions for CDKA in Theorem 3.4.

**Theorem 3.4.** *Consider CDKA of the form in Eq. (15) where $\boldsymbol{A}^{(i)}$ is initialized with Kaiming initialization (He et al., 2015), and $\boldsymbol{B}^{(i)}$ is initialized to zero. Then the gradient norm of CDKA is irrelevant to $r_1$, $r_2$ and $r$ if and only if $\lambda_{r_1, r_2, r} \in \Theta\left(\frac{1}{\sqrt{r \cdot r_2}}\right)$.*

Proof is referred to Appendix D. Following Theorem 3.4, we set the scaling factor as

$$\lambda_{r_1, r_2, r} = \frac{\alpha}{\sqrt{r \cdot r_2}}, \qquad (16)$$

*Table 9.* Performance of fine-tuning T5-Base model on the GLUE benchmark. Results marked with (*) are obtained from our reimplementation, using hyperparameters aligned with those reported in the original papers. All other results are sourced from prior works (He et al., 2025; Zhang et al., 2025).

| Method | Params(M) | MNLI | SST-2 | CoLA | QNLI | MRPC | Average |
|---|---|---|---|---|---|---|---|
| Full | 226 | $86.33_{\pm0.00}$ | $94.75_{\pm0.21}$ | $80.70_{\pm0.24}$ | $93.19_{\pm0.22}$ | $84.56_{\pm0.73}$ | 87.91 |
| LoRA | 3.24 | $85.30_{\pm0.04}$ | $94.04_{\pm0.11}$ | $69.35_{\pm0.05}$ | $92.96_{\pm0.09}$ | $68.38_{\pm0.01}$ | 82.08 |
| PiSSA | 3.24 | $85.75_{\pm0.07}$ | $94.07_{\pm0.06}$ | $74.27_{\pm0.39}$ | $93.15_{\pm0.14}$ | $76.31_{\pm0.51}$ | 84.71 |
| rsLoRA | 3.24 | $85.73_{\pm0.10}$ | $94.19_{\pm0.23}$ | $72.32_{\pm1.12}$ | $93.12_{\pm0.09}$ | $52.86_{\pm2.27}$ | 79.64 |
| LoRA+ | 3.24 | $85.81_{\pm0.09}$ | $93.85_{\pm0.24}$ | $77.53_{\pm0.20}$ | $93.14_{\pm0.03}$ | $74.43_{\pm1.39}$ | 84.95 |
| DoRA | 3.24 | $85.67_{\pm0.09}$ | $94.04_{\pm0.53}$ | $72.04_{\pm0.94}$ | $93.04_{\pm0.06}$ | $68.08_{\pm0.51}$ | 82.57 |
| AdaLoRA | 4.86 | $85.45_{\pm0.11}$ | $93.69_{\pm0.20}$ | $69.16_{\pm0.24}$ | $91.66_{\pm0.05}$ | $68.14_{\pm0.28}$ | 81.62 |
| GoRA | 3.05 | $\mathbf{85.91}_{\pm0.02}$ | $\mathbf{94.68}_{\pm0.43}$ | $79.86_{\pm0.35}$ | $\underline{93.27}_{\pm0.08}$ | $\underline{86.10}_{\pm0.20}$ | $\underline{87.96}$ |
| LoRA-GA | 3.24 | $85.70_{\pm0.09}$ | $94.11_{\pm0.18}$ | $80.57_{\pm0.20}$ | $93.18_{\pm0.06}$ | $85.29_{\pm0.24}$ | 87.77 |
| LoRA-One | 3.24 | $\underline{85.89}_{\pm0.08}$ | $\underline{94.53}_{\pm0.13}$ | $\mathbf{82.04}_{\pm0.22}$ | $\mathbf{93.37}_{\pm0.02}$ | $\mathbf{87.83}_{\pm0.37}$ | $\mathbf{88.73}$ |
| KronA* | **0.41** | $85.05_{\pm0.06}$ | $93.12_{\pm0.23}$ | $68.62_{\pm0.14}$ | $92.66_{\pm0.06}$ | $82.43_{\pm0.99}$ | 84.38 |
| CDKA (Ours) | **0.41** | $85.23_{\pm0.03}$ | $94.15_{\pm0.43}$ | $\underline{80.82}_{\pm0.10}$ | $92.87_{\pm0.04}$ | $83.44_{\pm0.67}$ | 87.30 |

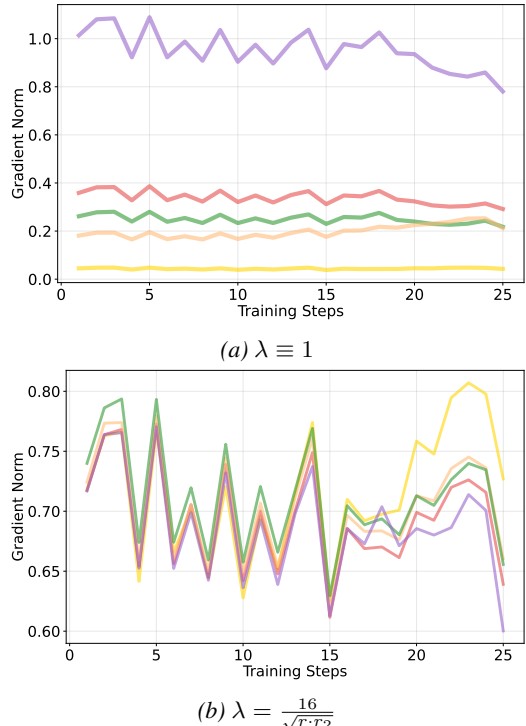

*(a)* $\lambda \equiv 1$

*(b)* $\lambda = \frac{16}{\sqrt{r \cdot r_2}}$

*Figure 1.* Gradient norm during training with (a) $\lambda \equiv 1$ and (b) $\lambda = \frac{16}{\sqrt{r \cdot r_2}}$ under different $r_1$, $r_2$ and $r$. Identical colors indicate identical component configurations. Our method maintains gradient norms at the same scale across different component configurations.

where $\alpha$ is a tunable hyperparameter. As shown in Figure 1b, for a fixed $\alpha = 16$, our method maintains gradient norms at the same scale across different component configurations, thereby ensuring training stability and consistency. See Appendix F for more results.

**Initialization strategies.** In the above analysis, we consistently adopt random initialization for component $\boldsymbol{A}^{(i)}$ while initializing $\boldsymbol{B}^{(i)}$ to zero. To justify this design choice, we investigate the impact of different initialization strategies on CDKA. We fix $r = 2$, $r_1 = 2$, $r_2 = 8$, and $\lambda \equiv 16$ to ensure fair comparisons. The results are summarized in Table 8, where Ku and Kn denote Kaiming uniform and Kaiming normal initialization (He et al., 2015), respectively. We observe that random initialization of $\boldsymbol{A}^{(i)}$ consistently outperforms random initialization of $\boldsymbol{B}^{(i)}$, confirming the effectiveness of our initialization design.

## 4. Experiments

In this section, we conduct experiments to evaluate the effectiveness of CDKA across a wide range of NLP and CV tasks. We begin by assessing its Natural Language Understanding (NLU) capability on a subset of GLUE dataset (Wang et al., 2018). To evaluate the capability of CDKA in Natural Language Generation (NLG) tasks, we evaluate CDKA on mathematical reasoning and code generation tasks. To further evaluate the effectiveness of CDKA on different modalities, we conduct experiments on seven image classification tasks with the CLIP-ViT-B/16 (Radford et al., 2021) model. All experiments are conducted under the same number of training epochs, with results reported as the mean and standard deviation over three random seeds. Further details on the hyperparameter settings are provided in Appendix A.

**Baselines.** We compare CDKA against a wide range of standard baselines, including LoRA (Hu et al., 2022), PiSSA (Meng et al., 2024), rsLoRA (Kalajdzievski, 2023), LoRA+ (Hayou et al., 2024), DoRA (Liu et al., 2024), AdaLoRA (Zhang et al., 2023), GoRA (He et al., 2025),

*Table 10.* Performance of fine-tuning LLaMA-2-7B. Results marked with (*) are obtained from our reimplementation. All other results are sourced from He et al. (2025).

| Method | GSM8k | HumanEval |
|---|---|---|
| Full | $59.36_{\pm0.85}$ | $35.31_{\pm2.13}$ |
| LoRA | $42.08_{\pm0.04}$ | $14.76_{\pm0.17}$ |
| PiSSA | $44.54_{\pm0.27}$ | $16.02_{\pm0.17}$ |
| rsLoRA | $45.62_{\pm0.10}$ | $16.01_{\pm0.79}$ |
| LoRA+ | $52.11_{\pm0.62}$ | $18.17_{\pm0.52}$ |
| DoRA | $53.07_{\pm0.75}$ | $19.75_{\pm0.41}$ |
| AdaLoRA | $50.72_{\pm1.39}$ | $17.80_{\pm0.44}$ |
| GoRA | $54.04_{\pm0.22}$ | $\mathbf{24.80}_{\pm1.04}$ |
| LoRA-GA | $53.60_{\pm0.30}$ | $19.81_{\pm1.46}$ |
| LoRA-One* | $\underline{55.40}_{\pm0.37}$ | $20.73_{\pm1.00}$ |
| KronA* | $49.00_{\pm0.41}$ | $17.21_{\pm2.01}$ |
| CDKA (Ours) | $\mathbf{56.71}_{\pm0.38}$ | $\underline{24.59}_{\pm2.74}$ |

*Table 11.* Performance of fine-tuning LLaMA-3.1-8B. Results marked with (*) are obtained from our reimplementation. All other results are sourced from He et al. (2025).

| Method | GSM8k |
|---|---|
| Full | $73.69_{\pm0.28}$ |
| LoRA | $67.78_{\pm1.25}$ |
| GoRA | $\underline{72.91}_{\pm0.76}$ |
| KronA* | $68.11_{\pm0.38}$ |
| CDKA (Ours) | $\mathbf{73.74}_{\pm0.42}$ |

*Table 12.* Performance of fine-tuning Qwen on MetaMathQA. Results with (*) are obtained from our reimplementation.

| Method | Qwen-3-0.6B | Qwen-3-8B |
|---|---|---|
| LoRA-One* | $\underline{64.77}_{\pm0.59}$ | $85.98_{\pm0.32}$ |
| KronA* | $60.85_{\pm0.13}$ | $85.06_{\pm0.67}$ |
| CDKA (Ours) | $\mathbf{65.83}_{\pm0.37}$ | $\mathbf{86.56}_{\pm0.20}$ |

LoRA-GA (Wang et al., 2024), LoRA-One (Zhang et al., 2025), LoRA-Pro (Wang et al., 2025) and KronA (Edalati et al., 2022). We exclude MoKA (Sadeghi et al., 2025) and MoKA-MoE (Yu et al., 2025) since they introduce substantial computational overhead. For the results on NLG tasks, we reproduce LoRA-One to ensure a consistent inference protocol. For KronA, we set $r_1 = r_2 = 64$ and $r = 1$ with $\lambda = 16$ to improve its performance.

### 4.1. Experiments on Natural Language Understanding

**Implementation Details.** To evaluate the capability of CDKA on NLU tasks, we follow the common experimental settings adopted in prior works (Wang et al., 2024; He et al., 2025; Zhang et al., 2025) and fine-tune the T5-Base model (Raffel et al., 2020) on five selected subsets (MNLI, SST-2, CoLA, QNLI, MRPC) in the GLUE benchmark (Wang et al., 2018). We report accuracy on the corresponding validation sets. For the component configuration of CDKA, we set $r_1 = 3$, $r_2 = 3$ and $r = 1$.

**Results.** As shown in Table 9, CDKA achieves competitive performance across all five tasks using only 12.5% of the trainable parameters compared to LoRA-One, without introducing additional training overhead. Compared to KronA, CDKA yields an average score improvement of 2.92 percentage points under the same parameter budget. Notably, CDKA requires only 0.18% of the trainable parameters to approach the performance of full fine-tuning, further highlighting the effectiveness of our approach.

### 4.2. Experiments on Natural Language Generation

**Implementation Details.** To evaluate the capability of CDKA in Natural Language Generation (NLG) tasks, we fine-tune the LLaMA-2-7B (Touvron et al., 2023) model on mathematical reasoning and code generation tasks. For mathematical reasoning, we train the model on a 100K subset of MetaMathQA (Yu et al., 2024) and evaluate on the test set of GSM8k (Cobbe et al., 2021). Performance is measured using the Exact Match (EM) metric. For code generation, we train the model on a 100K subset of Code-FeedBack (Zheng et al., 2024) and evaluate on HumanEval (Chen, 2021). Performance is measured using the PASS@1 metric. For the component configuration of CDKA, we set $r_1 = 2$, $r_2 = 2$, $r = 8$ for MetaMathQA and $r_1 = 2$, $r_2 = 8$, $r = 4$ for Code-FeedBack.

**Results.** As shown in Table 10, CDKA demonstrates consistent improvements across large-scale experiments. In particular, CDKA achieves state-of-the-art performance on mathematical reasoning tasks, outperforming LoRA-One by 1.31 percentage points. On code generation tasks, CDKA attains the second-best performance. Compared to KronA, CDKA achieves improvements of 7.71 and 4.94 percentage points respectively, highlighting the effectiveness of our method. To further investigate the adaptability of CDKA across different backbone models, we fine-tune LLaMA-3.1-8B (Grattafiori et al., 2024), Qwen-3-0.6B and Qwen-3-8B (Yang et al., 2025) on mathematical reasoning tasks, with the results reported in Table 11 and Table 12. Consistent with the results on LLaMA-2-7B, CDKA again achieves state-of-the-art performance, demonstrating the robustness and strong generalization ability of our approach across different backbone models.

### 4.3. Experiments on Image Classification

**Implementation Details.** To further evaluate the effectiveness of CDKA across different modalities, we conduct

*Table 13.* Performance of fine-tuning CLIP-ViT-B/16 model on seven image classification tasks. Results marked with (*) are obtained from our reimplementation. All other results are sourced from He et al. (2025).

| Method | Cars | DTD | EuroSAT | GTSRB | RESISC45 | SUN397 | SVHN | Average |
|---|---|---|---|---|---|---|---|---|
| Zero-shot | $63.75$ | $44.39$ | $42.22$ | $35.22$ | $56.46$ | $62.56$ | $15.53$ | $45.73$ |
| Full | $84.23_{\pm0.06}$ | $77.44_{\pm0.19}$ | $\underline{98.09}_{\pm0.03}$ | $94.31_{\pm0.28}$ | $93.95_{\pm0.00}$ | $75.35_{\pm0.10}$ | $93.04_{\pm0.18}$ | $88.06$ |
| LoRA | $72.81_{\pm0.13}$ | $73.92_{\pm0.38}$ | $96.93_{\pm0.07}$ | $92.40_{\pm0.16}$ | $90.03_{\pm0.14}$ | $70.12_{\pm0.18}$ | $88.02_{\pm0.07}$ | $83.46$ |
| rsLoRA | $82.38_{\pm0.20}$ | $78.03_{\pm0.76}$ | $98.06_{\pm0.08}$ | $95.04_{\pm0.11}$ | $93.96_{\pm0.18}$ | $75.38_{\pm0.24}$ | $92.74_{\pm0.18}$ | $87.94$ |
| LoRA+ | $72.87_{\pm0.18}$ | $74.07_{\pm0.45}$ | $97.18_{\pm0.07}$ | $92.40_{\pm0.17}$ | $90.23_{\pm0.18}$ | $70.17_{\pm0.15}$ | $88.08_{\pm0.05}$ | $83.57$ |
| DoRA | $73.72_{\pm0.06}$ | $73.72_{\pm0.33}$ | $96.95_{\pm0.01}$ | $92.38_{\pm0.08}$ | $90.32_{\pm0.08}$ | $70.20_{\pm0.16}$ | $88.23_{\pm0.05}$ | $83.48$ |
| LoRA-GA | $\underline{85.18}_{\pm0.41}$ | $77.50_{\pm0.12}$ | $98.05_{\pm0.27}$ | $95.28_{\pm0.10}$ | $94.33_{\pm0.19}$ | $75.44_{\pm0.06}$ | $93.68_{\pm0.35}$ | $88.51$ |
| LoRA-Pro | $\mathbf{85.87}_{\pm0.08}$ | $\mathbf{78.64}_{\pm0.25}$ | $98.46_{\pm0.03}$ | $95.66_{\pm0.05}$ | $94.75_{\pm0.21}$ | $\underline{76.42}_{\pm0.14}$ | $94.63_{\pm0.20}$ | $89.20$ |
| LoRA-One* | $82.75_{\pm0.23}$ | $78.42_{\pm0.35}$ | $\mathbf{99.03}_{\pm0.05}$ | $\underline{98.55}_{\pm0.19}$ | $\underline{95.58}_{\pm0.11}$ | $75.38_{\pm0.16}$ | $\underline{97.26}_{\pm0.06}$ | $\underline{89.57}$ |
| KronA* | $74.71_{\pm0.12}$ | $63.95_{\pm0.78}$ | $98.38_{\pm0.06}$ | $96.20_{\pm0.15}$ | $92.60_{\pm0.14}$ | $71.98_{\pm0.07}$ | $96.78_{\pm0.00}$ | $84.94$ |
| CDKA (Ours) | $84.35_{\pm0.14}$ | $\underline{78.53}_{\pm0.15}$ | $\mathbf{99.14}_{\pm0.04}$ | $\mathbf{98.65}_{\pm0.19}$ | $\mathbf{96.00}_{\pm0.16}$ | $\mathbf{76.42}_{\pm0.02}$ | $\mathbf{97.48}_{\pm0.00}$ | $\mathbf{90.08}$ |

*Table 14.* Robustness of CDKA under fixed component configurations. For CLIP-ViT-B/16, we report the average performance across seven datasets.

| Method | LLaMA-2-7B | Qwen-3-0.6B | Qwen-3-8B | CLIP-ViT-B/16 |
|---|---|---|---|---|
| LoRA-One | $\underline{55.40}_{\pm0.37}$ | $\underline{64.77}_{\pm0.59}$ | $\mathbf{85.98}_{\pm0.32}$ | $\underline{89.57}$ |
| KronA | $49.00_{\pm0.41}$ | $60.85_{\pm0.13}$ | $85.06_{\pm0.67}$ | $84.94$ |
| CDKA (Ours) | $\mathbf{56.48}_{\pm0.12}$ | $\mathbf{65.23}_{\pm0.19}$ | $\underline{86.35}_{\pm0.10}$ | $\mathbf{90.00}$ |

*Table 15.* Computational Costs of CDKA on MetaMathQA.

| Base Model | Method | Time Cost | Memory Cost |
|---|---|---|---|
| LLaMA-2-7B | LoRA | 7h32min29s | 18125MB |
| | CDKA | 7h49min31s | 18137MB |
| LLaMA-3.1-8B | LoRA | 5h40min32s | 23503MB |
| | CDKA | 5h48min03s | 23507MB |

experiments on image classification tasks. Specifically, we fine-tune CLIP-ViT-B/16 (Radford et al., 2021) on seven datasets, including Stanford Cars (Krause et al., 2013), DTD (Cimpoi et al., 2014), EuroSAT (Helber et al., 2019), GTSRB (Houben et al., 2013), RESISC45 (Cheng et al., 2017), SUN397 (Xiao et al., 2010), and SVHN (Netzer et al., 2011), and report the corresponding test accuracy. The classifier is constructed using prompts like "a photo of a class". We set $r_1 = 2$, $r_2 = 16$, and $r = 2$ for CDKA.

**Results.** As shown in Table 13, CDKA achieves consistent improvements across all tasks and state-of-the-art average accuracy. Notably, CDKA improves the average score by 5.14 percentage points over KronA and 0.51 percentage points over LoRA-One, demonstrating the strong generalization ability of our approach across different modalities.

### 4.4. Robustness of CDKA

To examine the robustness of CDKA, we conduct experiments using a "default" component configuration with $r_1 = 2$, $r_2 = 8$, and $r = 4$. This configuration follows our proposed design principles with a small $r_1$, a large $r_2$, and a moderate $r$. We evaluate it under diverse settings, including mathematical reasoning with LLaMA-2-7B, Qwen-3-0.6B, and Qwen-3-8B, as well as image classification with CLIP-ViT-B/16. As shown in Table 14, although alternative configurations may yield marginal improvements in specific cases, this simple "default" configuration consistently outperforms LoRA-One, the state-of-the-art LoRA variant,

and KronA. These results demonstrate the robustness of our design guidelines and the adaptability of CDKA across models and modalities.

### 4.5. Computational Costs

To avoid explicitly computing the Kronecker product $\boldsymbol{B}^{(i)} \otimes \boldsymbol{A}^{(i)}$, we adopt an equivalent reformulation following previous work (Edalati et al., 2022), which is expressed by:

$$(\boldsymbol{B}^{(i)} \otimes \boldsymbol{A}^{(i)})\boldsymbol{x} = \text{vec}\left(\boldsymbol{A}^{(i)} \boldsymbol{X} (\boldsymbol{B}^{(i)})^{\top}\right), \quad (17)$$

where $\boldsymbol{X} \in \mathbb{R}^{\frac{d_{\text{in}}}{r_2} \times r_2}$ is obtained by reshaping the input vector $\boldsymbol{x}$. This reformulation substantially reduces computational overhead. To verify the efficiency of CDKA, we evaluate its computational cost on a single RTX 4090 GPU. As shown in Table 15, the training time and memory consumption of CDKA are nearly identical to those of standard LoRA under the same trainable parameters, making CDKA computationally efficient in practice.

## 5. Conclusion and Limitations

In this paper, we perform a fine-grained analysis of how the dimensions and number of Kronecker components influence the performance of Kronecker adapters. We propose CDKA and provide practical guidelines for deployment. Experiments across diverse NLP and CV tasks demonstrate the effectiveness of CDKA. For simplicity, our theoretical analysis mainly focuses on linear settings, leaving a deeper nonlinear analysis of Kronecker adapters for future work.

## Impact Statement

This paper provides a detailed analysis of Kronecker adapters and derives principles for improving their performance. The target of this paper is to advance the field of Machine Learning. There might be some potential societal consequences of our work, none of which we feel must be specifically highlighted here.

## Acknowledgements

Z. Ling and Z. Liao would like to acknowledge the National Key Research and Development Program of China (No. 2025YFA1018600). Z. Ling would also like to acknowledge the National Natural Science Foundation of China (NSFC-62406119) and the Natural Science Foundation of Hubei Province (2024AFB074). Z. Liao would also like to acknowledge the National Natural Science Foundation of China (NSFC-12571561) and the Fundamental Research Support Program of HUST (2025BRSXB0004). F. Zhou would like to acknowledge the National Natural Science Foundation of China (NSFC-62576346), the MOE Project of Key Research Institute of Humanities and Social Sciences (22JJD110001), the Fundamental Research Funds for the Central Universities, the Research Funds of Renmin University of China (24XNKJ13), and the Beijing Advanced Innovation Center for Future Blockchain and Privacy Computing. R. C. Qiu would like to acknowledge the National Natural Science Foundation of China (NSFC-12141107), the Key Research and Development Program of Wuhan (2024050702030100), and the Key Research and Development Program of Guangxi (GuiKe-AB21196034).

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

## A. Experimental Details

We fine-tune all linear layers except for the language head for T5-Base, LLaMA-2-7B, Qwen-3-0.6B and Qwen-3-8B. For LLaMA-3.1-8B, we fine-tune all linear layers in the attention modules. For CLIP-ViT-B/16, we fine-tune all linear layers in the visual backbone. Other hyperparameters for the experiments are summarized in Table 16.

*Table 16.* Hyperparameters for CDKA on different models.

| Model | LR | Warmup | Optimizer | Betas | Weight Decay | Batch Size | $\alpha$ |
|-------|-----|--------|-----------|-------|--------------|------------|----------|
| T5-Base | 2e−3 | 0.03 | AdamW | (0.9, 0.999) | 0 | 32 | 16 |
| LLaMA-2-7B | 2e−4 | 0.03 | AdamW | (0.9, 0.999) | 0 | 32 | 64 |
| LLaMA-3.1-8B | 2e−4 | 0.03 | AdamW | (0.9, 0.999) | 5e−4 | 64 | 64 |
| Qwen-3-0.6B | 2e−4 | 0.03 | AdamW | (0.9, 0.999) | 0 | 32 | 32 |
| Qwen-3-8B | 2e−4 | 0.03 | AdamW | (0.9, 0.999) | 0 | 32 | 32 |
| CLIP-ViT-B/16 | 1e−4 | 0.03 | AdamW | (0.9, 0.999) | 0.01 | 64 | 32 |

## B. The Alignment of $\widetilde{A}_t$

**Theorem B.1** (The top-$r^*$ left singular subspace alignment between $\widetilde{A}_t$ and $\widetilde{G}_0$.). *Under the settings described in Section 3.2, we consider random Gaussian initialization for $\widetilde{A}_0$ with $[\widetilde{A}_0]_{ij} \sim \mathcal{N}(0, \alpha^2)$ and zero initialization for $\widetilde{B}_0$ with $[\widetilde{B}_0]_{ij} = 0$ and*

$$
\alpha \leq \begin{cases} \left( \dfrac{\theta \xi \sqrt{r_2}}{24 r \sqrt{r_1 d_{in}}} \right)^{\frac{3\kappa}{2}} \sqrt{\dfrac{\sigma_1(\widetilde{G}_0) r_2}{94.5 \sqrt{r} r_1 d_{in}}}, & if\ r^* \leq r < 2r^*, \\[2em] \left( \dfrac{\theta \sqrt{r_2}}{24 \sqrt{r_1 d_{in}}} \right)^{\frac{3\kappa}{2}} \sqrt{\dfrac{\sigma_1(\widetilde{G}_0) r_2}{94.5 \sqrt{r} r_1 d_{in}}}, & if\ r \geq 2r^*. \end{cases}
$$

*where $\kappa$ is the condition number of $\widetilde{G}_0$. Then if we run gradient descent for $t_A^*$ steps on the Kronecker adapter with:*

$$
t_A^* \lesssim \begin{cases} \dfrac{\ln\left( \frac{24 r \sqrt{r_1 d_{in}}}{\theta \xi \sqrt{r2}} \right)}{\ln\left( 1 + \eta \sigma_{r^*}(\widetilde{G}_0) \right)}, & if\ r^* \leq r < 2r^*, \\[2em] \dfrac{\ln\left( \frac{24 \sqrt{r_1 d_{in}}}{\theta \sqrt{r2}} \right)}{\ln\left( 1 + \eta \sigma_{r^*}(\widetilde{G}_0) \right)}, & if\ r \geq 2r^*, \end{cases}
$$

*we have the following alignment for $\forall \theta \in (0, 1)$:*

$$
\|\boldsymbol{U}_{r^*,\perp}^\top(\widetilde{G}_0) \boldsymbol{U}_{r^*}(\widetilde{A}_{t_A^*})\|_2 \leq \theta, \tag{18}
$$

*with probability at least*

$$
\begin{cases} 1 - C_1 \exp(-d_{in} \frac{r_1}{r_2}) - (C_2 \xi)^{r - r^* + 1} - C_3 \exp(-r) - C \exp(-N), & if\ r^* \leq r < 2r^*, \\ 1 - C_4 \exp(-d_{in} \frac{r_1}{r_2}) - C_5 \exp(-r) - C \exp(-N), & if\ r \geq 2r^*. \end{cases}
$$

*for some universal constants $C$, $C_1$, $C_2$, $C_3$, $C_4$, $C_5$.*

*Proof.* We divide the proof into three parts. First, we derive the dynamics of the Kronecker components $\widetilde{A}_t$ and $\widetilde{B}_t$. Next, we establish an error bound between the linearized dynamics of the Kronecker components and the original dynamics. Finally, we combine these results to characterize the alignment between $\widetilde{A}_t$ and $\widetilde{G}_0$.

**Part 1: The dynamic of Kronecker components**

Under the settings described in Section 3.2, we first derive the update rules of $\widetilde{A}_t$ and $\widetilde{B}_t$ under gradient descent. Combining Eq. (7) and Eq. (8), the iteration of a single component $A_t^{(i)}$ and $B_t^{(i)}$ can be expressed as follows:

$$\text{vec}(A_{t+1}^{(i)}) = \text{vec}(A_t^{(i)}) - \eta \, \text{Kreshape}\left(\frac{1}{N}\left((W_0 + \sum_{i=1}^{r} B^{(i)} \otimes A^{(i)})X - Y\right)X^\top\right) \text{vec}(B_t^{(i)}),$$

$$\text{vec}(B_{t+1}^{(i)}) = \text{vec}(B_t^{(i)}) - \eta \, \text{Kreshape}^\top\left(\frac{1}{N}\left((W_0 + \sum_{i=1}^{r} B_t^{(i)} \otimes A_t^{(i)})X - Y\right)X^\top\right) \text{vec}(A_t^{(i)}). \tag{19}$$

As a result, we can derive the following iteration for $\widetilde{A}_t = [\text{vec}(A_t^1), \cdots, \text{vec}(A_t^r)]$ and $\widetilde{B}_t = [\text{vec}(B_t^1), \cdots, \text{vec}(B_t^r)]$:

$$\widetilde{A}_{t+1} = \widetilde{A}_t + \eta \widetilde{G}_0 \widetilde{B}_t - \eta \, \text{Kreshape}\left(\frac{1}{N}\sum_{i=1}^{r} B_t^{(i)} \otimes A_t^{(i)} X X^\top\right)\widetilde{B}_t,$$

$$\widetilde{B}_{t+1} = \widetilde{B}_t + \eta \widetilde{G}_0^\top \widetilde{A}_t - \eta \, \text{Kreshape}^\top\left(\frac{1}{N}\sum_{i=1}^{r} B_t^{(i)} \otimes A_t^{(i)} X X^\top\right)\widetilde{A}_t, \tag{20}$$

since we have $\widetilde{G}_0 = \text{Kreshape}(G_0) = \text{Kreshape}\left(\frac{1}{N}(Y - W_0 X)X^\top\right)$ according to Eq. (10).

Let $Z_t = \begin{bmatrix} \widetilde{A}_t \\ \widetilde{B}_t \end{bmatrix}$, then we can rewrite the above iteration to:

$$Z_{t+1} = H Z_t - \hat{E}_{t+1}, \tag{21}$$

where

$$H = \begin{bmatrix} I_{\frac{r_1}{r_2} d_{\text{in}}} & \eta \widetilde{G}_0 \\ \eta \widetilde{G}_0^\top & I_{\frac{r_2}{r_1} d_{\text{out}}} \end{bmatrix}, \tag{22}$$

denotes the time-independent linear part and

$$\hat{E}_{t+1} = \eta \begin{bmatrix} 0 & \text{Kreshape}\left(\frac{1}{N}\sum_{i=1}^{r} B_t^{(i)} \otimes A_t^{(i)} X X^\top\right) \\ \text{Kreshape}^\top\left(\frac{1}{N}\sum_{i=1}^{r} B_t^{(i)} \otimes A_t^{(i)} X X^\top\right) & 0 \end{bmatrix} \begin{bmatrix} \widetilde{A}_t \\ \widetilde{B}_t \end{bmatrix}, \tag{23}$$

represents the nonlinear component of the update.

**Part 2: The error bound of the linear approximation**

We focus on the dynamics of the linear part in Eq. (21), namely,

$$Z_{t+1}^{lin} = H Z_t^{lin}, \tag{24}$$

since it is highly correlated with $\widetilde{G}_0$. We present the closed-form characterization of the linear dynamics of $Z_t^{lin}$, as stated in Lemma B.2.

**Lemma B.2** (Linear dynamic of the Kronecker adapter. Adapted from Lemma C.5 in Zhang et al. (2025).). *Under the setting described in Section 3.2, the dynamic of $Z_t^{lin}$ in Eq. (24) is given by*

$$\widetilde{A}_t^{lin} = P_t^A \widetilde{A}_0 = \frac{1}{2}\widetilde{U}\left((I_{\frac{r_1}{r_2} d_{in}} + \eta \widetilde{S})^t + (I_{\frac{r_1}{r_2} d_{in}} - \eta \widetilde{S})^t\right)\widetilde{U}^\top \widetilde{A}_0,$$

$$\widetilde{B}_t^{lin} = P_t^B \widetilde{A}_0 = \frac{1}{2}\widetilde{V}\left((I_{\frac{r_1}{r_2} d_{in}} + \eta \widetilde{S})^t - (I_{\frac{r_1}{r_2} d_{in}} - \eta \widetilde{S})^t\right)\widetilde{U}^\top \widetilde{A}_0, \tag{25}$$

*where $\widetilde{G}_0 = \widetilde{U}\widetilde{S}\widetilde{V}$ is the SVD of $\widetilde{G}_0$.*

We next derive the error between the true iteration $Z_t$ and its linear approximation $Z_t^{lin}$, which is given by

$$E_t = Z_t - Z_t^{lin} = -\sum_{i=1}^{t} H^{t-i}\hat{E}_i. \tag{26}$$

We formalize this result in Theorem B.3.

**Theorem B.3** (The error bound of the linear approximation). *Under the setting described in Section 3.2, consider the following time period*

$$t \le t^{lin} = \frac{\ln \left( \frac{\sigma_1(\widetilde{\boldsymbol{G}}_0)}{10.5\sqrt{r}\|\widetilde{\boldsymbol{A}}_0\|_2^2} \right)}{3 \ln \left( 1 + \eta\sigma_1(\widetilde{\boldsymbol{G}}_0) \right)}. \tag{27}$$

*Then the error in Eq. (26) is controlled by*

$$\|\boldsymbol{E}_t\|_2 \le \|\widetilde{\boldsymbol{A}}_0\|_2, \tag{28}$$

*with probability at least $1 - 2C\exp(-N)$ for a universal constant $C$.*

*Proof.* We prove this by induction. For $t = 0$, we have

$$\|\boldsymbol{E}_0\|_2 = 0 \le \|\widetilde{\boldsymbol{A}}_0\|_2.$$

For $t \ge 1$, we assume Eq. (28) holds for $t - 1$. Through Lemma B.2, we can derive the following bound for $\|\widetilde{\boldsymbol{A}}_{t-1}\|_2$:

$$\begin{aligned}
\|\widetilde{\boldsymbol{A}}_{t-1}\|_2 &\le \|\widetilde{\boldsymbol{A}}_{t-1}^{lin}\|_2 + \|\boldsymbol{E}_{t-1}\|_2 \\
&\le (1 + \eta\sigma_1(\widetilde{\boldsymbol{G}}_0))^{t-1}\|\widetilde{\boldsymbol{A}}_0\|_2 + \|\boldsymbol{E}_{t-1}\|_2.
\end{aligned}$$

Similarly, $\|\widetilde{\boldsymbol{B}}_{t-1}\|_2$ is bounded by

$$\begin{aligned}
\|\widetilde{\boldsymbol{B}}_{t-1}\|_2 &\le \|\widetilde{\boldsymbol{B}}_{t-1}^{lin}\|_2 + \|\boldsymbol{E}_{t-1}\|_2 \\
&\le \frac{1}{2}(1 + \eta\sigma_1(\widetilde{\boldsymbol{G}}_0))^{t-1}\|\widetilde{\boldsymbol{A}}_0\|_2 + \|\boldsymbol{E}_{t-1}\|_2.
\end{aligned}$$

Then with probability at least $1 - 2C\exp(-N\epsilon^2)$ for a universal constant $C$, we have:

$$\begin{aligned}
\|\hat{\boldsymbol{E}}_t\|_2 &\le \eta\| \text{Kreshape}\,\big(\frac{1}{N}\sum_{i=1}^{r}\boldsymbol{B}_{t-1}^{(i)} \otimes \boldsymbol{A}_{t-1}^{(i)}\boldsymbol{X}\boldsymbol{X}^\top\big)\widetilde{\boldsymbol{B}}_{t-1}\|_2 + \eta\|\text{Kreshape}^\top\big(\frac{1}{N}\sum_{i=1}^{r}\boldsymbol{B}_{t-1}^{(i)} \otimes \boldsymbol{A}_{t-1}^{(i)}\boldsymbol{X}\boldsymbol{X}^\top\big)\widetilde{\boldsymbol{A}}_{t-1}\|_2 \\
&\le \eta\|\frac{1}{N}\sum_{i=1}^{r}\boldsymbol{B}_{t-1}^{(i)} \otimes \boldsymbol{A}_{t-1}^{(i)}\boldsymbol{X}\boldsymbol{X}^\top\|_F\big(\|\widetilde{\boldsymbol{A}}_{t-1}\|_2 + \|\widetilde{\boldsymbol{B}}_{t-1}\|_2\big) \\
&\le \eta\|\frac{1}{N}\boldsymbol{X}\boldsymbol{X}^\top\|_2\|\sum_{i=1}^{r}\boldsymbol{B}_{t-1}^{(i)} \otimes \boldsymbol{A}_{t-1}^{(i)}\|_F\big(\|\widetilde{\boldsymbol{A}}_{t-1}\|_2 + \|\widetilde{\boldsymbol{B}}_{t-1}\|_2\big) \\
&\le (1 + \epsilon)\eta\sqrt{r}\|\widetilde{\boldsymbol{A}}_{t-1}\|_2\|\widetilde{\boldsymbol{B}}_{t-1}\|_2\big(\|\widetilde{\boldsymbol{A}}_{t-1}\|_2 + \|\widetilde{\boldsymbol{B}}_{t-1}\|_2\big) \\
&\le (1 + \epsilon)\eta\sqrt{r}\big((1 + \eta\sigma_1(\widetilde{\boldsymbol{G}}_0))^{t-1}\|\widetilde{\boldsymbol{A}}_0\|_2 + \|\boldsymbol{E}_{t-1}\|_2\big)\big(\frac{1}{2}(1 + \eta\sigma_1(\widetilde{\boldsymbol{G}}_0))^{t-1}\|\widetilde{\boldsymbol{A}}_0\|_2 + \|\boldsymbol{E}_{t-1}\|_2\big) \\
&\quad \big(\frac{3}{2}(1 + \eta\sigma_1(\widetilde{\boldsymbol{G}}_0))^{t-1}\|\widetilde{\boldsymbol{A}}_0\|_2 + 2\|\boldsymbol{E}_{t-1}\|_2\big) \\
&\le 10.5(1 + \epsilon)\eta\sqrt{r}(1 + \eta\sigma_1(\widetilde{\boldsymbol{G}}_0))^{3t-3}\|\widetilde{\boldsymbol{A}}_0\|_2^3.
\end{aligned}$$

As a result, we can get the following bound on $\|\boldsymbol{E}_t\|_2$, namely:

$$\begin{aligned}
\|\boldsymbol{E}_t\|_2 &= \|\sum_{i=1}^{t}\boldsymbol{H}^{t-i}\hat{\boldsymbol{E}}_i\|_2 \\
&\le \sum_{i=1}^{t}\|\boldsymbol{H}\|_2^{t-i}\|\hat{\boldsymbol{E}}_i\|_2 \\
&\le 10.5(1 + \epsilon)\eta\sqrt{r}\|\widetilde{\boldsymbol{A}}_0\|_2^3\sum_{i=1}^{t}(1 + \eta\sigma_1(\widetilde{\boldsymbol{G}}_0))^{t+2i-3} \\
&\le 5.25(1 + \epsilon)\sqrt{r}(1 + \eta\sigma_1(\widetilde{\boldsymbol{G}}_0))^{3t}\frac{\|\widetilde{\boldsymbol{A}}_0\|_2^3}{\sigma_1(\widetilde{\boldsymbol{G}}_0)}.
\end{aligned}$$

By taking $\epsilon = 1$, when $t \leq t^{lin} = \dfrac{\ln\left(\frac{\sigma_1(\widetilde{G}_0)}{10.5\sqrt{r}\|\widetilde{A}_0\|_2^2}\right)}{3\ln\left(1+\eta\sigma_1(\widetilde{G}_0)\right)}$, we have

$$\|E_t\|_2 \leq \|\widetilde{A}_0\|_2,$$

with probability at least $1 - 2C\exp(-N)$ for a universal constant $C$, which proves the claim. $\qquad\square$

**Part 3: The alignment between $\widetilde{A}_t$ and $\widetilde{G}_0$**

Building upon the above results, we are now ready to derive the alignment between $\widetilde{A}_t$ and $\widetilde{G}_0$ through Lemma B.4.

**Lemma B.4** (Adapted from Lemma C.8 in Zhang et al. (2025).)**.** *Under the settings described in Section 3.2. If we run gradient descent for $t_A^*$ steps on the Kronecker adapter with*

$$t_A^* \leq \frac{\ln\left(\frac{8\|\widetilde{A}_0\|_2}{\theta\,\sigma_{\min}\left(U_{r^*}^\top(P_{t_A^*}^A)\widetilde{A}_0\right)}\right)}{\ln\left(1+\eta\sigma_{r^*}(\widetilde{G}_0)\right)}, \tag{29}$$

*and $t_A^* \leq t^{lin}$, then for $\forall\theta \in (0,1)$, we have the following alignment:*

$$\|U_{r^*,\perp}^\top(\widetilde{G}_0)U_{r^*}(\widetilde{A}_{t_A^*})\|_2 \leq \theta, \tag{30}$$

*with probability at least $1 - 2C\exp(-N)$ for a universal constant $C$.*

Through Lemma B.4, we know that the alignment can be achieved when $t_A^* \leq t^{lin}$, which indicates that

$$\frac{\ln\left(\frac{8\|\widetilde{A}_0\|_2}{\theta\,\sigma_{\min}\left(U_{r^*}^\top(P_{t_A^*}^A)\widetilde{A}_0\right)}\right)}{\ln\left(1+\eta\sigma_{r^*}(\widetilde{G}_0)\right)} \leq \frac{\ln\left(\frac{\sigma_1(\widetilde{G}_0)}{10.5\sqrt{r}\|\widetilde{A}_0\|_2^2}\right)}{3\ln\left(1+\eta\sigma_1(\widetilde{G}_0)\right)}.$$

**Case 1.** $r^* \leq r < 2r^*$. Using Lemma E.3 and Lemma E.4, we have the following bound with probability at least $1 - C_1\exp(-d_{\mathrm{in}}\frac{r_1}{r_2}) - (C_2\xi)^{r-r^*+1} - C_3\exp(-r) - C\exp(-N)$ for some universal constants $C, C_1, C_2, C_3$:

$$t_A^* \lesssim \frac{\ln\left(\frac{24r\sqrt{r_1 d_{\mathrm{in}}}}{\theta\xi\sqrt{r2}}\right)}{\ln\left(1+\eta\sigma_{r^*}(\widetilde{G}_0)\right)},$$

if the variance of $[\widetilde{A}_0]_{ij}$ satisfies:

$$\frac{\ln\left(\frac{24r\sqrt{r_1 d_{\mathrm{in}}}}{\theta\xi\sqrt{r2}}\right)}{\ln\left(1+\eta\sigma_{r^*}(\widetilde{G}_0)\right)} \leq \frac{\ln\left(\frac{\sigma_1(\widetilde{G}_0)r_2}{94.5\sqrt{r}r_1 d_{\mathrm{in}}\alpha^2}\right)}{3\ln\left(1+\eta\sigma_1(\widetilde{G}_0)\right)},$$

which indicates that

$$\alpha \leq \left(\frac{\theta\xi\sqrt{r_2}}{24r\sqrt{r_1 d_{\mathrm{in}}}}\right)^{\frac{3\kappa}{2}}\sqrt{\frac{\sigma_1(\widetilde{G}_0)r_2}{94.5\sqrt{r}r_1 d_{\mathrm{in}}}}.$$

**Case 2.** $r \geq 2r^*$. Using Lemma E.3 and Lemma E.4, we have the following bound with probability at least $1 - C_4\exp(-d_{\mathrm{in}}\frac{r_1}{r_2}) - C_5\exp(-r) - C\exp(-N)$ for some universal constants $C, C_4, C_5$:

$$t_A^* \lesssim \frac{\ln\left(\frac{24\sqrt{r_1 d_{\mathrm{in}}}}{\theta\sqrt{r2}}\right)}{\ln\left(1+\eta\sigma_{r^*}(\widetilde{G}_0)\right)},$$

if the variance of $[\widetilde{A}_0]_{ij}$ satisfies:

$$\frac{\ln\left(\frac{24\sqrt{r_1 d_{\mathrm{in}}}}{\theta\sqrt{r2}}\right)}{\ln\left(1 + \eta\sigma_{r^*}(\widetilde{G}_0)\right)} \leq \frac{\ln\left(\frac{\sigma_1(\widetilde{G}_0)r_2}{94.5\sqrt{r}r_1 d_{\mathrm{in}}\alpha^2}\right)}{3\ln\left(1 + \eta\sigma_1(\widetilde{G}_0)\right)},$$

which indicates that

$$\alpha \leq \left(\frac{\theta\sqrt{r_2}}{24\sqrt{r_1 d_{\mathrm{in}}}}\right)^{\frac{3\kappa}{2}}\sqrt{\frac{\sigma_1(\widetilde{G}_0)r_2}{94.5\sqrt{r}r_1 d_{\mathrm{in}}}},$$

which proves the claim. □

# C. The Alignment of $\widetilde{B}_t$

Building on the concept of Kronecker product singular value decomposition in Definition 2.1, we quantify the alignment between $\widetilde{B}_t$ and the gradient $G_0$ with:

$$\|V_{r^*,\perp}^\top(\widetilde{G}_0)V_{r^*}(\widetilde{B}_t^\top)\|_2, \tag{31}$$

where $r^*$ is the rank of $\widetilde{G}_0 = \mathrm{Kreshape}(G_0)$ and $\widetilde{B}_t = [\mathrm{vec}(B_t^{(1)}), \cdots, \mathrm{vec}(B_t^{(r)})]$. Then we can derive the alignment $\widetilde{B}_t$ and $\widetilde{G}_0$ in Theorem C.1.

**Theorem C.1** (The top-$r^*$ right singular subspace alignment between $\widetilde{B}_t$ and $\widetilde{G}_0$.). *Under the settings described in Section 3.2, we consider random Gaussian initialization for $\widetilde{A}_0$ with $[\widetilde{A}_0]_{ij} \sim \mathcal{N}(0, \alpha^2)$ and zero initialization for $\widetilde{B}_0$ with $[\widetilde{B}_0]_{ij} = 0$ and*

$$\alpha \leq \begin{cases} \exp\left(-\frac{9\kappa r\sqrt{r_1 d_{in}}}{\eta\theta\xi\sqrt{r2}}\right)\sqrt{\frac{\sigma_1(\widetilde{G}_0)r_2}{94.5\sqrt{r}r_1 d_{in}}}, & \text{if } r^* \leq r < 2r^*, \\ \exp\left(-\frac{9\kappa\sqrt{r_1 d_{in}}}{\eta\theta\sqrt{r2}}\right)\sqrt{\frac{\sigma_1(\widetilde{G}_0)r_2}{94.5\sqrt{r}r_1 d_{in}}}, & \text{if } r \geq 2r^*. \end{cases}$$

*where $\kappa$ is the condition number of $\widetilde{G}_0$. Then if we run gradient descent for $t_B^*$ steps on the Kronecker adapter with:*

$$t_B^* \lesssim \begin{cases} \dfrac{6r\sqrt{r_1 d_{in}}}{\theta\xi\sqrt{r2} \cdot \eta\sigma_{r^*}(\widetilde{G}_0)}, & \text{if } r^* \leq r < 2r^*, \\ \dfrac{6r\sqrt{r_1 d_{in}}}{\theta\sqrt{r2} \cdot \eta\sigma_{r^*}(\widetilde{G}_0)}, & \text{if } r \geq 2r^*, \end{cases}$$

*we have the following alignment for $\forall\theta \in (0,1)$:*

$$\|V_{r^*,\perp}^\top(\widetilde{G}_0)V_{r^*}(\widetilde{B}_{t_B^*}^\top)\|_2 \leq \theta, \tag{32}$$

*with probability at least*

$$\begin{cases} 1 - C_1\exp(-d_{in}\frac{r_1}{r_2}) - (C_2\xi)^{r-r^*+1} - C_3\exp(-r) - C\exp(-N), & \text{if } r^* \leq r < 2r^*, \\ 1 - C_4\exp(-d_{in}\frac{r_1}{r_2}) - C_5\exp(-r) - C\exp(-N), & \text{if } r \geq 2r^*. \end{cases}$$

*for some universal constants $C, C_1, C_2, C_3, C_4, C_5$.*

*Proof.* The proof follows a similar strategy to that of Theorem B.1. In particular, the first two steps are identical, and therefore we start directly from the third step. To derive an upper bound on $\|V_{r^*,\perp}^\top(\widetilde{G}_0)V_{r^*}(\widetilde{B}_t^\top)\|_2$, we invoke the following lemma, whose assumption is inherited from the necessary condition of Wedin's $\sin\theta$ theorem (Wedin, 1972).

**Lemma C.2** (Adapted from Lemma 8.3 in Stöger & Soltanolkotabi (2021).). *We assume that*

$$\sigma_{r^*+1}(P_t^B)\|\widetilde{A}_0\|_2 + \|E_t\|_2 < \sigma_{r^*}(P_t^B)\sigma_{min}(V_{r^*}^\top(P_t^B)\widetilde{A}_0), \tag{33}$$

*then the following three inequalities hold:*

$$\sigma_{r^*}(P_t^B\widetilde{A}_0 + E_t) \geq \sigma_{r^*}(P_t^B)\sigma_{min}(V_{r^*}^\top(P_t^B)\widetilde{A}_0) - \|E_t\|_2 \tag{34}$$

$$\sigma_{r^*+1}(\boldsymbol{P}_t^B \widetilde{\boldsymbol{A}}_0 + \boldsymbol{E}_t) \leq \sigma_{r^*+1}(\boldsymbol{P}_t^B)\|\widetilde{\boldsymbol{A}}_0\|_2 + \|\boldsymbol{E}_t\|_2 \tag{35}$$

$$\|\boldsymbol{V}_{r^*,\perp}^\top(\widetilde{\boldsymbol{G}}_0)\boldsymbol{V}_{r^*}(\widetilde{\boldsymbol{B}}_t^\top)\|_2 \leq \frac{\sigma_{r^*+1}(\boldsymbol{P}_t^B)\|\widetilde{\boldsymbol{A}}_0\|_2 + \|\boldsymbol{E}_t\|_2}{\sigma_{r^*}(\boldsymbol{P}_t^B)\sigma_{min}(\boldsymbol{V}_{r^*}^\top(\boldsymbol{P}_t^B)\widetilde{\boldsymbol{A}}_0) - \sigma_{r^*+1}(\boldsymbol{P}_t^B)\|\widetilde{\boldsymbol{A}}_0\|_2 - \|\boldsymbol{E}_t\|_2}. \tag{36}$$

Building on Lemma C.2, we can derive the alignment between $\widetilde{\boldsymbol{B}}_t$ and $\widetilde{\boldsymbol{G}}_0$ in Theorem C.3.

**Theorem C.3.** *Under the settings described in Section 3.2. If we run gradient descent for $t_B^*$ steps on the Kronecker adapter with*

$$t_B^* \leq \frac{2\|\widetilde{\boldsymbol{A}}_0\|_2}{\theta\sigma_{min}(\boldsymbol{V}_{r^*}^\top(\boldsymbol{P}_{t_B^*}^B)\widetilde{\boldsymbol{A}}_0)\eta\sigma_{r^*}(\widetilde{\boldsymbol{G}})}, \tag{37}$$

*and $t_B^* \leq t^{lin}$, then for $\forall\theta \in (0,1)$, we have the following alignment:*

$$\|\boldsymbol{V}_{r^*,\perp}^\top(\widetilde{\boldsymbol{G}})\boldsymbol{V}_{r^*}(\widetilde{\boldsymbol{B}}_{t_B^*}^\top)\|_2 \leq \theta, \tag{38}$$

*with probability at least $1 - 2C\exp(-N)$ for a universal constant $C$.*

*Proof.* Following Lemma C.2, the following holds for $\forall\theta \in (0,1)$

$$\|\boldsymbol{V}_{r^*,\perp}^\top(\widetilde{\boldsymbol{G}})\boldsymbol{V}_{r^*}(\widetilde{\boldsymbol{B}}_t^\top)\|_2 \leq \theta,$$

when

$$\frac{\sigma_{r^*+1}(\boldsymbol{P}_t^B)\|\widetilde{\boldsymbol{A}}_0\|_2 + \|\boldsymbol{E}_t\|_2}{\sigma_{r^*}(\boldsymbol{P}_t^B)\sigma_{min}(\boldsymbol{V}_{r^*}^\top(\boldsymbol{P}_t^B)\widetilde{\boldsymbol{A}}_0)} \leq \frac{\theta}{2}.$$

Using the result in Lemma B.2 and Theorem B.3, then under the assumption $t_B^* \leq t^{lin}$, we can derive the following upper bound for $t_B^*$:

$$t_B^* \leq \frac{2\|\widetilde{\boldsymbol{A}}_0\|_2}{\theta\sigma_{min}(\boldsymbol{V}_{r^*}^\top(\boldsymbol{P}_{t_B^*}^B)\widetilde{\boldsymbol{A}}_0)\eta\sigma_{r^*}(\widetilde{\boldsymbol{G}})},$$

which proves the claim. $\qquad\square$

Through Lemma B.4, we know that the alignment can be achieved when $t_B^* \leq t^{lin}$, which indicates that

$$\frac{2\|\widetilde{\boldsymbol{A}}_0\|_2}{\theta\sigma_{min}(\boldsymbol{V}_{r^*}^\top(\boldsymbol{P}_{t_B^*}^B)\widetilde{\boldsymbol{A}}_0)\eta\sigma_{r^*}(\widetilde{\boldsymbol{G}})} \leq \frac{\ln\left(\frac{\sigma_1(\widetilde{\boldsymbol{G}}_0)}{10.5\sqrt{r}\|\widetilde{\boldsymbol{A}}_0\|_2^2}\right)}{3\ln\left(1 + \eta\sigma_1(\widetilde{\boldsymbol{G}}_0)\right)}.$$

**Case 1.** $r^* \leq r < 2r^*$. Using Lemma E.3 and Lemma E.4, we have the following bound with probability at least $1 - C_1\exp(-d_{in}\frac{r_1}{r_2}) - (C_2\xi)^{r-r^*+1} - C_3\exp(-r) - C\exp(-N)$ for some universal constants $C, C_1, C_2, C_3$:

$$t_B^* \lesssim \frac{6r\sqrt{r_1 d_{in}}}{\theta\xi\sqrt{r2} \cdot \eta\sigma_{r^*}(\widetilde{\boldsymbol{G}}_0)},$$

if the variance of $[\widetilde{\boldsymbol{A}}_0]_{ij}$ satisfies:

$$\frac{6r\sqrt{r_1 d_{in}}}{\theta\xi\sqrt{r2} \cdot \eta\sigma_{r^*}(\widetilde{\boldsymbol{G}}_0)} \leq \frac{\ln\left(\frac{\sigma_1(\widetilde{\boldsymbol{G}}_0)r_2}{94.5\sqrt{r}r_1 d_{in}\alpha^2}\right)}{3\ln\left(1 + \eta\sigma_1(\widetilde{\boldsymbol{G}}_0)\right)},$$

which indicates that

$$\alpha \leq \exp\left(-\frac{9\kappa r\sqrt{r_1 d_{in}}}{\eta\theta\xi\sqrt{r2}}\right)\sqrt{\frac{\sigma_1(\widetilde{\boldsymbol{G}}_0)r_2}{94.5\sqrt{r}r_1 d_{in}}}.$$

**Case 2.** $r \geq 2r^*$. Using Lemma E.3 and Lemma E.4, we have the following bound with probability at least $1 - C_4 \exp(-d_{\text{in}}\frac{r_1}{r_2}) - C_5 \exp(-r) - C \exp(-N)$ for some universal constants $C, C_4, C_5$:

$$t_B^* \lesssim \frac{6r\sqrt{r_1 d_{\text{in}}}}{\theta\sqrt{r2} \cdot \eta\sigma_{r^*}(\widetilde{\boldsymbol{G}}_0)},$$

if the variance of $[\widetilde{\boldsymbol{A}}_0]_{ij}$ satisfies:

$$\frac{6\sqrt{r_1 d_{\text{in}}}}{\theta\sqrt{r2} \cdot \eta\sigma_{r^*}(\widetilde{\boldsymbol{G}}_0)} \leq \frac{\ln\left(\frac{\sigma_1(\widetilde{\boldsymbol{G}}_0)r_2}{94.5\sqrt{r}r_1 d_{\text{in}}\alpha^2}\right)}{3\ln\left(1 + \eta\sigma_1(\widetilde{\boldsymbol{G}}_0)\right)},$$

which indicates that

$$\alpha \leq \exp\left(-\frac{9\kappa\sqrt{r_1 d_{\text{in}}}}{\eta\theta\sqrt{r2}}\right)\sqrt{\frac{\sigma_1(\widetilde{\boldsymbol{G}}_0)r_2}{94.5\sqrt{r}r_1 d_{\text{in}}}},$$

which proves the claim. $\qquad\square$

## D. Proof of Theorem 3.4

We assume that the Kronecker adapter is trained using the loss function $\mathcal{L}_{\text{KA}}$, which is minimized via gradient descent with learning rate $\eta$. During the forward pass at the $t$-th iteration, given an input $\boldsymbol{x}_t$, the output of the Kronecker adapter is expressed as

$$\boldsymbol{y}_t = \lambda\sum_{i=1}^{r} \boldsymbol{B}_t^{(i)} \otimes \boldsymbol{A}_t^{(i)}\boldsymbol{x}_t. \tag{39}$$

Here, $\boldsymbol{A}_t^{(i)}$ and $\boldsymbol{B}_t^{(i)}$ denote the values of $\boldsymbol{A}^{(i)}$ and $\boldsymbol{B}^{(i)}$, respectively, after $t$ steps of gradient descent.

During backpropagation, given the gradient with respect to the output $\boldsymbol{v}_t = \frac{\partial\mathcal{L}_{\text{KA}}}{\partial\boldsymbol{y}_t}$, the gradient with respect to the input $\boldsymbol{x}_t$ is given by

$$\boldsymbol{g}_t = \frac{\partial\mathcal{L}_{\text{KA}}}{\partial\boldsymbol{x}_t} = \lambda\sum_{i=1}^{r}(\boldsymbol{B}_t^{(i)})^\top \otimes (\boldsymbol{A}_t^{(i)})^\top\boldsymbol{v}_t. \tag{40}$$

To ensure that the gradient norm of CDKA does not vary with changes in $r_1$, $r_2$, and $r$, the scales of $\boldsymbol{y}_t$ and $\boldsymbol{g}_t$ must be independent of $r_1$, $r_2$, and $r$. To this end, we first derive the update rules for $\boldsymbol{A}^{(i)}$ and $\boldsymbol{B}^{(i)}$:

$$\begin{aligned} \frac{\partial\mathcal{L}_{\text{KA}}}{\partial\boldsymbol{A}_t^{(i)}} &= \lambda\boldsymbol{V}_t\boldsymbol{B}_t^{(i)}\boldsymbol{X}_t^\top, \\ \frac{\partial\mathcal{L}_{\text{KA}}}{\partial\boldsymbol{B}_t^{(i)}} &= \lambda\boldsymbol{V}_t^\top\boldsymbol{A}_t^{(i)}\boldsymbol{X}_t, \end{aligned} \tag{41}$$

where $\boldsymbol{X}_t \in \mathbb{R}^{\frac{d_{\text{in}}}{r_2}\times r_2}$ is reshaped by $\boldsymbol{x}_t$ and $\boldsymbol{V}_t \in \mathbb{R}^{r_1\times\frac{d_{\text{out}}}{r_1}}$ is reshaped by $\boldsymbol{v}_t$. Under the assumption that $\boldsymbol{B}_0^{(i)}$ is initialized to zero, we can derive the following formulation for $\boldsymbol{A}_t^{(i)}$ and $\boldsymbol{B}_t^{(i)}$ by induction:

$$\boldsymbol{A}_t^{(i)} = \boldsymbol{A}_0^{(i)} + O(\lambda^2), \tag{42}$$

$$\boldsymbol{B}_t^{(i)} = -\eta\lambda\sum_{k=0}^{t-1}\boldsymbol{V}_k^\top\boldsymbol{A}_0^{(i)}\boldsymbol{X}_k + O(\lambda^2). \tag{43}$$

Since $\boldsymbol{A}_0^{(i)}$ is initialized using Kaiming initialization (He et al., 2015), the variance scale of $\boldsymbol{A}_0^{(i)}$ is $\Theta(r_2)$. Consequently, we obtain the following expressions for the scales of the output $\boldsymbol{y}_t$ and the input gradient $\boldsymbol{g}_t$:

$$\boldsymbol{y}_t = -\lambda^2\eta\sum_{i=1}^{r}\text{vec}(\boldsymbol{A}_0^{(i)}\boldsymbol{X}_t\sum_{k=0}^{t-1}\boldsymbol{X}_k^\top(\boldsymbol{A}_0^{(i)})^\top\boldsymbol{V}_k) + O(\lambda^3) \in \Theta(\lambda^2 rr_2), \tag{44}$$

$$\boldsymbol{g}_t = -\lambda^2 \eta \sum_{i=1}^{r} \text{vec}((\boldsymbol{A}_0^{(i)})^\top \boldsymbol{V}_t \sum_{k=0}^{t-1} \boldsymbol{V}_k^\top \boldsymbol{A}_0^{(i)} \boldsymbol{X}_k) + O(\lambda^3) \in \Theta(\lambda^2 r r_2). \tag{45}$$

As a result, the scales of $\boldsymbol{y}_t$ and $\boldsymbol{g}_t$ are independent of $r_1$, $r_2$ and $r$ if and only if

$$\lambda \in \Theta\Big(\frac{1}{\sqrt{r \cdot r_2}}\Big). \tag{46}$$

## E. Basic Definitions and Lemmas

In this section, we present some basic definitions and lemmas that are needed for our proof.

**Definition E.1.** $\text{Kreshape}(\cdot)$ is a function that reshapes a matrix

$$\boldsymbol{K} = \begin{bmatrix} \boldsymbol{K}_{1,1} & \cdots & \boldsymbol{K}_{1,r_2} \\ \vdots & \ddots & \vdots \\ \boldsymbol{K}_{\frac{d_{\text{out}}}{r_1},1} & \cdots & \boldsymbol{K}_{\frac{d_{\text{out}}}{r_1},r_2} \end{bmatrix}, \quad \boldsymbol{K}_{i,j} \in \mathbb{R}^{r_1 \times \frac{d_{\text{in}}}{r_2}}, \tag{47}$$

to:

$$\text{Kreshape}(\boldsymbol{K}) = [\text{vec}(\boldsymbol{K}_{1,1}), \cdots, \text{vec}(\boldsymbol{K}_{\frac{d_{\text{out}}}{r_1},r_2})]. \tag{48}$$

**Lemma E.2** (Adapted from Theorem 4.6.1 in Vershynin (2018).). *Let $\boldsymbol{X} \in \mathbb{R}^{d_{in} \times N}$ whose columns $\boldsymbol{x}_i$ are independent, mean zero, sub-gaussian isotropic random vectors, then we have*

$$\|\frac{1}{N}\boldsymbol{X}\boldsymbol{X}^\top - \boldsymbol{I}_{d_{in}}\|_2 \leq \epsilon, \tag{49}$$

*with probability at least $1 - 2C\exp(-N\epsilon^2)$ for a positive constant $C$.*

**Lemma E.3** (Adapted from Corollary 5.35 in Vershynin (2010).). *Let $\boldsymbol{A} \in \mathbb{R}^{d \times r}$ with $d > 2r$, whose entries are independent standard Gaussian random variables, then we have*

$$\|\boldsymbol{A}\|_2 \leq 3\sqrt{d}, \tag{50}$$

*with probability at least $1 - C\exp(-d)$ for a positive constant $C$.*

**Lemma E.4** (Adapted from Lemma E.3 in Zhang et al. (2025).). *Let $\boldsymbol{A} \in \mathbb{R}^{d \times r}$ with $d > 2r$, whose entries are independent standard Gaussian random variables and $\boldsymbol{U} \in \mathbb{R}^{d \times r^*}$ with orthonormal columns. If $r \geq 2r^*$, then we have*

$$\sigma_{\min}(\boldsymbol{U}^\top \boldsymbol{A}) \gtrsim 1, \tag{51}$$

*with probability at least $1 - C\exp(-r)$ for a positive constant $C$. If $r^* \leq r < 2r$, then we have*

$$\sigma_{\min}(\boldsymbol{U}^\top \boldsymbol{A}) \gtrsim \frac{\xi}{r}, \tag{52}$$

*with probability at least $1 - (C_1 \xi)^{r-r^*-1} - C_2 \exp(-r)$ for some positive constants $C_1$ and $C_2$.*

## F. Ablation Studies

**Scaling Factor.** To examine the performance gains of the stabilization scaling factor $\lambda$ versus the component design, we fine-tune LLaMA-2-7B on mathematical reasoning task. The detailed results are presented in Table 17. It can be observed that the performance gains of CDKA primarily stem from our component design, while the stabilization scaling factor further improves the performance under different component configurations. These results fully demonstrate the effectiveness of our method.

*Table 17.* Ablation study on the Stabilization Scaling Factor.

| Method | GSM8k |
|---|---|
| KronA | $49.00_{\pm0.41}$ |
| KronA + Stabilization Factor | $49.43_{\pm0.37}$ |
| KronA + Component Design | $\underline{55.62}_{\pm0.39}$ |
| KronA + Stabilization Factor + Component Design (CDKA) | $\mathbf{56.71}_{\pm0.38}$ |

**Robustness of Our Theoretical Principles.** To evaluate the robustness of our theoretical principles under different backbone models and modalities, we additionally fine-tune Qwen-3-0.6B on mathematical reasoning task and CLIP-ViT-B/16 on image classification task. The detailed results are presented in Table 18 to 20. It can be observed that these additional results further support our theoretical principles in practice, which demonstrate the robustness of our principles across different settings.

*Table 18.* CDKA with different $r_1$ for fixed $r_2$ and $r$. Increasing $r_1$ tends to degrade the performance of CDKA.

| $r_1$ | GSM8k(Qwen-3-0.6B) | Cars | DTD | EuroSAT | GTSRB | RESISC45 | SUN397 | SVHN | Average(CLIP-ViT-B/16) |
|---|---|---|---|---|---|---|---|---|---|
| 2 | $\mathbf{62.85}_{\pm0.38}$ | **78.31** | **71.12** | **98.70** | **97.81** | **94.29** | **73.69** | **96.91** | **89.69** |
| 8 | $61.68_{\pm0.04}$ | 73.11 | 61.28 | 98.63 | 96.42 | 91.87 | 70.90 | 96.74 | 88.70 |
| 32 | $62.02_{\pm0.35}$ | 71.81 | 52.82 | 98.07 | 95.19 | 90.95 | 68.08 | 96.65 | 81.94 |

*Table 19.* CDKA with different $r_2$ for fixed $r_1$ and $r$. Increasing $r_2$ consistently improves the performance of CDKA.

| $r_2$ | GSM8k(Qwen-3-0.6B) | Cars | DTD | EuroSAT | GTSRB | RESISC45 | SUN397 | SVHN | Average(CLIP-ViT-B/16) |
|---|---|---|---|---|---|---|---|---|---|
| 2 | $62.85_{\pm0.38}$ | 78.31 | 71.12 | 98.70 | 97.81 | 94.29 | 73.69 | 96.91 | 89.69 |
| 8 | $63.76_{\pm0.53}$ | 81.37 | 75.90 | 99.04 | **98.39** | 95.14 | 74.74 | 97.18 | 88.82 |
| 32 | $\mathbf{65.58}_{\pm0.08}$ | **84.42** | **78.67** | **99.19** | 98.30 | **96.08** | **76.22** | **97.18** | **90.01** |

*Table 20.* CDKA with different $r$ for fixed $r_1$ and $r_2$. Increasing $r$ does not lead to a sustained improvement in the performance of CDKA.

| $r$ | GSM8k(Qwen-3-0.6B) | Cars | DTD | EuroSAT | GTSRB | RESISC45 | SUN397 | SVHN | Average(CLIP-ViT-B/16) |
|---|---|---|---|---|---|---|---|---|---|
| 2 | $64.25_{\pm1.78}$ | 79.87 | 75.00 | 98.85 | 98.06 | 94.76 | 74.02 | 97.00 | 88.22 |
| 8 | $64.90_{\pm0.38}$ | 83.17 | 78.03 | **99.11** | 98.50 | 95.94 | 75.66 | 97.44 | 89.69 |
| 32 | $\mathbf{65.24}_{\pm0.49}$ | 84.65 | **81.06** | 98.96 | 98.61 | 95.89 | **76.35** | **97.70** | 90.46 |
| 128 | $64.26_{\pm0.12}$ | **86.39** | 80.59 | 99.11 | **98.94** | **96.22** | 76.05 | 97.62 | **90.70** |
| 512 | $61.94_{\pm0.91}$ | 77.73 | 73.56 | 98.19 | 98.73 | 92.51 | 67.96 | 97.03 | 86.53 |

**Robustness of Our Proposed Guidelines.** To evaluate the robustness of our theoretical principles under different backbone models and modalities, we fine-tune LLaMA-3-70B on mathematical reasoning task and CLIP-ViT-B/16 on image classification task. The detailed results are presented in Table 21 to 24. It can be observed that our guidelines exhibit consistent effectiveness across different settings, demonstrating the robustness of our method.

*Table 21.* LLaMA-3-70B results with different $r_1$ and $r_2$ under the same parameter budget.

| $r_1, r_2, r$ | GSM8k |
|---|---|
| $2, 2, 8$ | **84.23** |
| $8, 8, 8$ | 84.00 |
| $64, 64, 8$ | 83.62 |
| $2, 16, 2$ | **85.22** |
| $8, 64, 2$ | 84.76 |

*Table 22.* LLaMA-3-70B results with different $r$ and $r_2$ under the same parameter budget.

| $r_1, r_2, r$ | GSM8k |
|---|---|
| $2, 2, 2$ | 83.09 |
| $2, 8, 1$ | **83.40** |
| $2, 2, 8$ | 84.23 |
| $2, 16, 2$ | **85.22** |

*Table 23.* CLIP-ViT-B/16 results with different $r_1$ and $r_2$ under the same parameter budget.

| $r_1, r_2, r$ | Cars | DTD | EuroSAT | GTSRB | RESISC45 | SUN397 | SVHN | Average |
|---|---|---|---|---|---|---|---|---|
| $2, 2, 8$ | **83.17** | **78.03** | **99.11** | **98.50** | **95.94** | **75.66** | **97.44** | **89.69** |
| $8, 8, 8$ | 81.06 | 75.21 | 99.07 | 97.99 | 95.35 | 74.93 | 97.28 | 88.70 |
| $32, 32, 8$ | 79.77 | 74.26 | 98.78 | 97.71 | 94.41 | 74.73 | 97.01 | 88.09 |

*Table 24.* CLIP-ViT-B/16 results with different $r$ and $r_2$ under the same parameter budget.

| $r_1, r_2, r$ | Cars | DTD | EuroSAT | GTSRB | RESISC45 | SUN397 | SVHN | Average |
|---|---|---|---|---|---|---|---|---|
| $2, 2, 8$ | 83.17 | 78.03 | **99.11** | **98.50** | 95.94 | 75.66 | 97.44 | 89.69 |
| $2, 16, 2$ | 84.39 | 78.51 | 99.00 | 98.47 | 96.16 | 76.25 | 97.45 | 90.03 |
| $2, 2, 32$ | 84.65 | **81.06** | 98.96 | 98.61 | 95.89 | 76.35 | **97.70** | 90.46 |
| $2, 16, 8$ | **86.23** | 79.57 | **99.33** | **98.88** | **96.25** | **76.75** | 97.48 | **90.64** |

