# OpenReview forum: "Diving into Kronecker Adapters: Component Design Matters"
_ICML.cc/2026/Conference — ICML 2026 regular_

### Official Review · Reviewer_cZKd · 2026-03-02

**Soundness:** 4
**Presentation:** 3
**Significance:** 3
**Originality:** 4
**Overall Recommendation:** 5
**Confidence:** 3

**Summary:**

This paper studies Kronecker-product adapters written as sums of Kronecker components (Eq. (2)) and argues that empirical performance depends not only on the attainable rank, but also on how that rank is realized through the component configuration (r1, r2, r). To support this, the paper analyzes the relationship between component design and alignment with full fine-tuning through the Kronecker SVD perspective, then proposes Component Designed Kronecker Adapters (CDKA), together with parameter-budget-aware design guidelines and a scaling-based stabilization strategy. The empirical study covers NLU, mathematical reasoning, and code generation, and shows that CDKA improves substantially over earlier Kronecker adapter variants while remaining competitive with strong PEFT baselines.

**Compliance With Llm Reviewing Policy:**

Affirmed.

**Final Justification:**

The paper is strong in originality, technical soundness, and empirical support, and the rebuttal clearly addressed my main concerns about evaluation matching and positioning against calibration baselines, which strengthened my assessment and led me to raise my recommendation.

**Key Questions For Authors:**

1. Under a fixed parameter budget, what default procedure would you recommend for choosing (r1, r2, r) on a new task or backbone? In practice, how many validation trials are usually needed before reaching a good configuration?
2. Which part of the analysis is most predictive of empirical performance in practice? For example, can the authors show more directly how the theoretical preference over component configurations correlates with the measured results across multiple settings?
3. The current experiments cover several NLP tasks and more than one backbone family. How stable is the proposed design guidance when moving to substantially different model scales or architectures?
4. Could the authors provide a minimal reproducible recipe for the stabilization strategy, including the recommended scaling, initialization, optimizer settings, and any additional implementation details that are important in practice?

**Limitations:**

Unclear

**Strengths And Weaknesses:**

Strengths

1. The paper isolates a concrete and practically relevant question for Kronecker adapters: how the choice of component dimensions and the number of components affects performance beyond the nominal attainable rank. The inconsistency example in Table 2 makes this point clearly and gives the paper a well-defined focus.
2. The analysis is anchored in a general Kronecker adapter formulation (Eq. (2)) together with explicit parameter and rank relations (Eqs. (5) and (6)). This makes the discussion of budget–capacity trade-offs more transparent and gives a reasonable basis for the proposed design principles.
3. The paper does more than critique existing constrained configurations. It turns the analysis into a concrete method, CDKA, and provides practical guidance for choosing component configurations under a fixed parameter budget. This makes the contribution more useful than a purely diagnostic study.

Weaknesses

1. While the paper shows convincingly that attainable rank alone is not sufficient to explain performance, the resulting guidance for choosing (r1, r2, r) on a new task is still somewhat heuristic. The practical story would be stronger with a simpler default selection rule that requires very little task-specific tuning.
2. The connection between the theoretical analysis and the final recommended configurations is suggestive, but not yet fully closed. In particular, it would help to see more direct evidence that the proposed principles reliably predict strong configurations across tasks and backbones without repeated empirical search.
3. The stabilization strategy is motivated, and its main ingredients are given, including the scaling rule and initialization choice, but the practical recipe is somewhat spread across the main text and appendix. A more compact end-to-end description would make reproduction easier.

---

> ### Author Rebuttal · Authors · 2026-03-31
>
> We thank the reviewer for your careful consideration. We greatly appreciate the positive comments and address your concerns below.
>
> **Robustness of Our Theoretical Principles in Practice**: Following your suggestions, we evaluate the robustness of our theoretical principles under different settings, including fine-tuning Qwen3-0.6B-Base on mathematical reasoning task and CLIP-ViT-B/16 on image classification task. The detailed results are presented in the following tables.  It can be observed that these additional results further support our theoretical principles in practice, which demonstrate the robustness of our principles across different settings.
>
> Table 1. CDKA with different $r_1$ for fixed $r_2$ and $r$. Increasing $r_1$ tends to degrade the performance of CDKA.
> | $r_1$ | GSM8k(Qwen3-0.6B-Base) | Cars |DTD |EuroSAT |GTSRB |RESISC45 |SUN397 |SVHN| Average(CLIP-ViT-B/16) |
> |:---                     |:---:|:---:|:---:|:---:|:---:|:---:|:---:|:---:|:---:|
> |2                      |$\mathbf{62.85}_{\pm0.38}$|**78.31**|**71.12**|**98.70**|**97.81**|**94.29**|**73.69**|**96.91**|**89.69**|
> |8                      |$61.68_{\pm0.04}$|73.11|61.28|98.63|96.42|91.87|70.90|96.74|88.70|
> |32                      |$62.02_{\pm0.35}$|71.81|52.82|98.07|95.19|90.95|68.08|96.65|81.94|
>
> Table 2. CDKA with different $r_2$ for fixed $r_1$ and $r$. Increasing $r_2$ consistently improves the performance of CDKA.
> | $r_2$ | GSM8k(Qwen3-0.6B-Base) | Cars |DTD |EuroSAT |GTSRB |RESISC45 |SUN397 |SVHN| Average(CLIP-ViT-B/16) |
> |:---                     |:---:|:---:|:---:|:---:|:---:|:---:|:---:|:---:|:---:|
> |2                      |$62.85_{\pm0.38}$|78.31|71.12|98.70|97.81|94.29|73.69|96.91|89.69|
> |8                      |$63.76_{\pm0.53}$|81.37|75.90|99.04|**98.39**|95.14|74.74|97.18|88.82|
> |32                      |$\mathbf{65.58}_{\pm 0.08}$|**84.42**|**78.67**|**99.19**|98.30|**96.08**|**76.22**|**97.18**|**90.01**|
>
> Table 3. CDKA with different $r$ for fixed $r_1$ and $r_2$. Increasing $r$ does not lead to a sustained improvement in the performance of CDKA.
> | $r$ | GSM8k(Qwen3-0.6B-Base) | Cars |DTD |EuroSAT |GTSRB |RESISC45 |SUN397 |SVHN| Average(CLIP-ViT-B/16) |
> |:---                     |:---:|:---:|:---:|:---:|:---:|:---:|:---:|:---:|:---:|
> |2                      |$64.25_{\pm1.78}$|79.87| 75.00|98.85|98.06|94.76|74.02|97.00|88.22|
> |8                      |$64.90_{\pm0.38}$|83.17|78.03|**99.11**|98.50|95.94|75.66|97.44|89.69|
> |32                      |$\mathbf{65.24}_{\pm0.49}$|84.65|**81.06**|98.96|98.61|95.89|**76.35**|**97.70**|90.46|
> |128                     |$64.26_{\pm0.12}$|**86.39**|80.59|99.11|**98.94**|**96.22**|76.05|97.62|**90.70**|
> |512                     |$61.94_{\pm0.91}$|77.73|73.56|98.19|98.73|92.51|67.96|97.03|86.53|
>
> **Robustness of Our Guidelines**: We direct the reviewer's attention to the "Results on Massive-scale LLMs and ViTs" section in our response to Reviewer Unc3. The detailed results are presented in Tables 1-4 in https://anonymous.4open.science/r/To-reviewer-cZKd-06C8. It can be observed that **fixing $r_1 = 2$** generally yields the best performance across different settings. For the choice of $r$ and $r_2$, we find that under a fixed parameter budget, it is sufficient to run **at most four validation trials**, including $r =1, 2, 4, 8$ with the corresponding $r_2$, to achieve the best performance, as we find increasing $r_2$ is more effective than increasing $r$ under sufficient parameter budget.
>
>
> **Implementation details** We have provided the implementation details in Table 13 of Appendix A in our original manuscript. For all experiments, we use Kaiming uniform initialization for $A$ and initialize $B$ to zero. For the experiments in this rebuttal, we uniformly adopt the following settings:
>
> | Optimizer | Betas| Learning Rate | Warmup | Batch Size | Weight Decay | $\alpha$(in the scaling factor)|
> |:---:                     |:---:|:---:|:---:|:---:|:---:|:---:|
> |AdamW| (0.9,0.999)|2e-4 | 0.03 | 64 | 0 | 32 |

---

> > ### Author Rebuttal · Reviewer_cZKd · 2026-04-01
> >
> > The rebuttal is helpful and addresses part of my concerns. In particular, the added results on Qwen3-0.6B-Base and CLIP-ViT-B/16 strengthen the empirical case that the proposed component-design principles remain useful across additional settings, and the added implementation details make the practical recipe clearer.
> >
> > That said, my main remaining concern is only partially resolved: the rebuttal provides more supporting experiments, but the connection between the theoretical preference over component configurations and the final recommended default choices is still not fully closed. In particular, I still think the paper would be stronger with more direct evidence that the theory reliably predicts strong configurations across tasks/backbones, rather than mainly providing post hoc empirical support.

---

> > > ### Author Response · Authors · 2026-04-03
> > >
> > > We thank the reviewer for the timely and helpful feedback. The choice of  $r_1$, $r_2$ and $r$ in KA, similar to the rank in LoRA,  are predefined hyperparameters, but their joint selection leads to a much larger design space. Without guidance, this would require expensive grid search. In contrast, our analysis provides qualitative but principled guidance, indicating a preference for a small $r_1$, a large $r_2$ and a moderate $r$, which substantially reduces the effective search space.
> > >
> > > Based on these insights,  we kindly note that even a very simple configuration such as  $r_1 = 2, r_2 =8, r =4$ performs well. Although alternative configurations may yield marginal improvements in specific settings, this simple configuration consistently outperforms LoRA-One, the SOTA LoRA variant, as well as KronA across a diverse range of settings, as demonstrated in the following tables.
> > >
> > > Table 1. Qwen3-0.6B-Base, Qwen3-8B-Base and LLaMA-2-7B on mathematical reasoning.
> > >
> > > | Method | Qwen3-0.6B-Base | Qwen3-8B-Base | LLaMA-2-7B |
> > > |:---                     |:---:|:---:|:---:|
> > > |LoRA-One                  |$64.41_{\pm0.35}$|$86.09_{\pm0.34}$|  $55.40_{\pm0.37}$|
> > > |KronA                     |  $60.77_{\pm0.04}$   | $84.61_{\pm0.08}$ |$49.00_{\pm0.41}$|
> > > |CDKA ($r_1 =2 , r_2 = 8, r =4$)               |$\mathbf{65.09}_{\pm0.04}$|$\mathbf{86.39}_{\pm 0.11}$| $\mathbf{56.48}_{\pm 0.12}$|
> > >
> > > Table 2. LLaMA-2-7B on commonsense reasoning.
> > >
> > > | Method | BoolQ | PIQA | SIQA | HellaSwag | WinoGrande | ARC-e | ARC-c | OBQA | Average |
> > > |:---|:---:|:---:|:---:|:---:|:---:|:---:|:---:|:---:|:---:|
> > > |LoRA-One |73.36|85.75|82.29|95.05|85.95|88.34| 75.17|86.40|84.04|
> > > |KronA | 67.22| 79.82 | 79.27 | 90.11 | 81.61| 81.23| 65.27 |79.20|77.97|
> > > |CDKA ($r_1 =2 , r_2 = 8, r =4$) | **74.07** | **86.18** | **82.34** | **95.29** | **86.50** | **88.93** | **75.26** | **87.20**|**84.47**|
> > >
> > > Table 3. CLIP-ViT-B/16 on image classification task.
> > > | Method | Cars |DTD |EuroSAT |GTSRB |RESISC45 |SUN397 |SVHN| Average |
> > > |:---                     |:---:|:---:|:---:|:---:|:---:|:---:|:---:|:---:|
> > > |LoRA-One                   |82.68|77.77|**99.22**|98.42|95.48|75.34|97.34|89.46|
> > > |KronA                     |74.75|63.24|98.52|96.12|92.84|72.12|96.77|84.91|
> > > |CDKA ($r_1 =2 , r_2 = 8, r =4$)                |**84.22**|**79.04**|99.00|**98.69**|**95.94**|**76.31**|**97.47**|**90.10**|
> > >
> > > We sincerely appreciate your overall positive assessment of our work. We will incorporate additional discussions and experimental results based on your suggestions in the final version

---

### Official Review · Reviewer_B4jt · 2026-03-09

**Soundness:** 3
**Presentation:** 3
**Significance:** 3
**Originality:** 3
**Overall Recommendation:** 5
**Confidence:** 4

**Summary:**

This paper discusses the best-practice of using Kronecker Adapters from the perspective of hyperparameter settings.  When fine-tuning large-scale models, LoRA is currently the most popular approach. It assumes that weight updates can be approximated by the product of two low-rank matrices, expressed as $W_0 + BA$. Kronecker adapters take this a step further by utilizing the Kronecker product to achieve high-rank updates. However, the authors discovered a counter-intuitive phenomenon in their experiments.  They proposed that simply increasing the maximum rank achievable by the adapter does not lead to stable performance gains and may even result in performance degradation.

The authors first analyze three core design principles:
- Increasing $r_1$ hurts performance. Larger values of $r_1$ typically lead to performance degradation of Kronecker adapters.
- Increasing $r_2$ consistently improves performance. Holding other variables constant, continuously increasing $r_2$ leads to stable improvements in model effectiveness.
- Increasing $r$ yields diminishing marginal returns. Increasing the number of components $r$ is effective initially, but once $r$ exceeds a certain threshold ($r^*$), performance stops improving and may even fluctuate or decline.

Based on these three principles, the authors provide a configuration formula, i.e., the CDKA Guidelines:
- Fix $r_1$: Set it to a minimal constant (e.g., 2 or 4).
- Search for the optimal $r$: Experiment with $r$ within a small range (typically between 2 and 8).
- Allocate excess parameters to $r_2$: If there is additional parameter budget, do not increase $r$ or $r_1$; instead, invest all remaining budget into increasing $r_2$.

**Compliance With Llm Reviewing Policy:**

Affirmed.

**Final Justification:**

My concerns have been adequately addressed. I think it is in general an interesting and technically solid paper. Discussing the best practice of LoKr may not seem to be so "novel" but I believe it will be a practical contribution to the PEFT community.

**Key Questions For Authors:**

1. Regarding the assumption of isotropic sub-Gaussian inputs in Section 3.2: have the authors conducted any empirical checks to see how much the alignment bound deviates when using real hidden states from a pre-trained LLM?
2. Could the authors clarify if the scaling factor $\lambda = \sqrt{\frac{\alpha}{r \cdot r_2}}$ remains optimal when the number of Kronecker components $n > 2$?
3. Please provide the missing legend for Figure 1 and define $U$ in Equation 11 to improve the clarity of the presentation.

**Limitations:**

The authors have not adequately discussed the limitations, and their societal impact statement is essentially a boilerplate dismissal.  I personally suggest the author to discuss the limitations of this work from the following points:
- The authors should explicitly state this as a limitation and discuss how this gap might affect the behavior of CDKA when applied to highly non-linear deep neural networks like Transformers.
-The empirical validation currently stops at the LLaMA-2, suggesting using new-generation models e.g., Qwen 3/3.5
- Provide a meaningful societal impact statement.

**Strengths And Weaknesses:**

### Soundness

 **Strengths:**
- In Theorem 3.2, the authors utilize KSVD and the Wedin sin $\theta$ theorem to derive the subspace alignment error bound of the adapter during the initial phase of training.
- The controlled experiments (Tables 3, 4, and 5), which independently vary $r_1, r_2$, and $r$, successfully validate the predictions made by the mathematical derivations.
- The adaptive scaling factor derived in Theorem 3.4 is well-justified and effectively addresses training stability.


**Weaknesses:**
- Section 3.2 relies on a simplified linear model assumption where input samples are assumed to follow an isotropic zero-mean sub-Gaussian distribution: $$\mathbf{x} \sim \mathcal{SG}(0, \sigma^2 \mathbf{I})$$


- Real-world LLMs are highly non-linear, and the distribution of real data is far from sub-Gaussian. I am concerned that these assumptions might hold less weight in complex non-linear components, such as the weight matrices $W$ following activation functions in MLPs or within Attention mechanisms. However, I generally consider this a reasonable simplification for mathematical tractability.



### Presentation

**Strengths:**
- The overall writing is fluent and easy to follow.
- The introduction to related work and the background on Kronecker adapters is well-structured and appropriate.


**Weaknesses:**
Some presentation details need further polishing:
- Fig 1 is missing a legend, making it difficult to interpret at a glance.
- The variable $U$ in Equation 11 should be explicitly defined for clarity.

### Significance

 **Strengths:**
- On NLU, math, and coding tasks, CDKA achieves comparable or superior performance to LoRA-One while using only 12.5% of the trainable parameters.
- This implies that under the same memory budget, the model can maintain a significantly higher effective rank.
- The proposed stable scaling factor $\lambda = \sqrt{\frac{\alpha}{r \cdot r_2}}$ effectively resolves fluctuations in the gradient norm.



### Originality

**Strengths:**
- The primary originality lies in establishing a "best practice" for Kronecker-based tuning. While integrating Kronecker products into fine-tuning is not entirely new e.g., KronA and MoKA, prior work often defaulted to $r_1 = r_2$ for simplicity or parameter compression.
- This paper identifies and proves that an asymmetric design—specifically, a very small $r_1$ coupled with a large $r_2$—is the optimal configuration for Kronecker adapters.

---

> ### Author Rebuttal · Authors · 2026-03-31
>
> We thank the reviewer for your careful consideration. We greatly appreciate the positive comments and address your concerns below.
>
> **Simplified Settings**: We thank the reviewer for noting that the linear and sub-guassian data assumptions are reasonable simplification for mathematical tractability. Under this simplified setting, which is standard in seminal LoRA-related theoretical works[1-3], a rigorous theoretical analysis of KAs has been lacking due to the intrinsic complexity of the Kronecker structure, while our work fills this gap. Moreover, our theoretical findings are supported by empirical results, suggesting that the insights remain relevant beyond the simplified assumptions.
>
> To further validate the effectiveness of our theoretical principles, we conduct experiments under different settings. Please see the next section for the detailed results.
>
> **Theoretical Principles in Real-World Settings** Following your suggestion, we further explore how the theoretical alignment behaves in real-world scenarios and how it varies with component design. For faster computation, we numerically evaluate the alignment between the subspace of $\tilde{G}$ and that of $\tilde{A}$, i.e., $\||U_{r*}^\top(\tilde{G}) U_{r*}(\tilde{A}) \||_2$ instead of metric used in our theory (the exact alignment between the complement of $\tilde{G}$ and $\tilde{A}$). We trace the behavior of $W_q$, $W_k$, $W_v$ in the attention modules and $W_u$, $W_d$ in the mlp modules. The result is evaluated on Qwen3-0.6B-Base with different $r, r_1, r_2$ after 500 training iterations. The detailed results are provided in Figures 1-3 in https://anonymous.4open.science/r/To-reviewer-B4jt-2E6D. It can be observed that the alignment in real-world LLMs is consistent with our theoretical findings, indicating the extensibility of our theory to real-world scenarios.
>
> To further evaluate how this alignment influences the practical performance of CDKA, we conduct experiments on more settings, including fine-tuning **Qwen3-0.6B-Base** on mathematical reasoning task and **CLIP-ViT-B/16** on image classification task. The detailed results are provided in Tables 1-3 in https://anonymous.4open.science/r/To-reviewer-B4jt-2E6D. It can be observed that these additional results further support our theoretical principles in practice, which demonstrate the robustness of our principles across different settings.
>
> **Scaling Factor with more Kronecker Components**: It is important to emphasize that the scaling factor $\lambda_{r_1, r_2, r} = \frac{\alpha}{\sqrt{r \cdot r_2}}$ already accounts for different choices of the number of Kronecker components, where we use $r$ to denote the number instead of $n$. We believe the misunderstanding mainly stems from the missing legend in Figure 1 in our original manuscipt, as the results in Figure 1 already include those with more Kronecker components. We will incorporate this in the revised manuscript.
>
> **Robustness of CDKA on More Models and Datasets**: Following your suggestion, we evaluate the performance of CDKA across different architectures and tasks, including **Qwen3-0.6B-Base** and **Qwen3-8B-Base** on mathematical reasoning and **CLIP-ViT-B/16** on image classification. We compare CDKA with LoRA-One, the state-of-the-art LoRA variant. The detailed results are presented in the following tables. It can be observed that **CDKA consistently outperforms LoRA-One across all settings**, demonstrating the robustness of our proposed method.
>
> Table 2. Qwen3-0.6B-Base and Qwen3-8B-Base on mathematical reasoning.
>
> | Method | Qwen3-0.6B-Base | Qwen3-8B-Base |
> |:---                     |:---:|:---:|
> |LoRA-One                  |$64.41_{\pm0.35}$|$86.09_{\pm0.34}$|
> |CDKA (Ours)               |$\mathbf{65.58}_{\pm 0.08}$|$\mathbf{86.70}_{\pm 0.04}$|
>
> Table 3. CLIP-ViT-B/16 on image classification task.
> | Method | Cars |DTD |EuroSAT |GTSRB |RESISC45 |SUN397 |SVHN| Average |
> |:---                     |:---:|:---:|:---:|:---:|:---:|:---:|:---:|:---:|
> |LoRA-One                   |82.68|77.77|**99.22**|98.42|95.48|75.34|97.34|89.46|
> |CDKA (Ours)                |**84.39**|**78.51**|99.00|**98.47**|**96.16**|**76.25**|**97.45**|**90.03**|
>
> **Presentation:** We thank the reviewer for the valuable suggestions regarding the missing legend in Figure 1 and the definition of $U$. We will incorporate these in the revised manuscript.
>
> [1] LoRA+: Efficient Low Rank Adaptation of Large Models ICML2024
>
> [2] Riemannian Preconditioned LoRA for Fine-Tuning Foundation Models ICML2024
>
> [3] LoRA-One: One-Step Full Gradient Could Suffice for Fine-Tuning Large Language Models, Provably and Efficiently ICML2025

---

> > ### Author Rebuttal · Reviewer_B4jt · 2026-04-02
> >
> > Thanks for the author's reply and I think it is in general an interesting paper. I will raise my score for supporting the acceptance of the paper. Best : ).

---

> > > ### Author Response · Authors · 2026-04-03
> > >
> > > We are glad that you find our work interesting. All the discussions and additional experiments will be included in the final version.

---

### Official Review · Reviewer_Unc3 · 2026-03-11

**Soundness:** 2
**Presentation:** 3
**Significance:** 3
**Originality:** 2
**Overall Recommendation:** 4
**Confidence:** 4

**Summary:**

This paper investigates the component design of Kronecker Adapters for parameter-efficient fine-tuning (PEFT). It reveals that the dimensions of the Kronecker components ($r_1$, $r_2$) and the number of components ($r$) significantly impact the adapter's expressive capacity and empirical performance. Supported by a theoretical analysis based on Kronecker Singular Value Decomposition (SVD), the authors propose Component Designed Kronecker Adapters (CDKA) along with parameter-budget-aware guidelines and a specific training stabilization strategy.

**Compliance With Llm Reviewing Policy:**

Affirmed.

**Final Justification:**

My concerns have been resolved, so I have raised my score to 4.

**Key Questions For Authors:**

* How robust are the proposed design guidelines when applied to substantially larger models (e.g., >70B parameters) or different architectures (e.g., Vision Transformers for image classification)?
* The theoretical analysis relies heavily on a simplified linear setting. Could you elaborate on how these linear assumptions might break down in deep, non-linear networks, and whether that affects your component design guidelines?
* Could authors provide an ablation study isolating the performance gains of the stabilization scaling factor $\lambda$ versus the pure component design ($r_1, r_2, r$) itself? How much of the final performance gain is strictly due to the scaling factor?

**Limitations:**

* The theoretical findings are constrained by linear approximations and specific initialization assumptions.
* The empirical evaluation does not cover extreme-scale models or cross-modal applications, limiting the proven universality of the method.

**Strengths And Weaknesses:**

### Strengths
* The paper provides a principled and exhaustive discussion on how hyperparameters ($r_1$, $r_2$, $r$) affect the performance of Kronecker Adapters, backed by solid theoretical alignment with full fine-tuning dynamics. The proposed CDKA, along with its stabilization scaling factor and initialization strategy, yields strong empirical results on various NLP tasks with a highly efficient parameter budget compared to baselines like LoRA and KronA.

### Weaknesses
* While the theoretical analysis using Kronecker SVD is comprehensive, the practical algorithmic contribution essentially boils down to heuristic guidelines for hyperparameter selection and a variance-based scaling factor. The underlying architectural framework remains largely identical to existing Kronecker adapter formulations.
* The empirical evaluation is strictly restricted to NLP tasks on mid-sized models (T5-Base, LLaMA-2-7B, LLaMA-3.1-8B). It remains unclear if these component design principles (e.g., keeping $r_1$ small) hold true for fundamentally different modalities (e.g., Vision Transformers) or massive-scale LLMs (e.g., LLaMA-3-70B or larger).
* The theoretical bounds heavily rely on a simplified linear setting and assume input samples are drawn from an isotropic, zero-mean sub-Gaussian distribution. This diverges significantly from the non-linear realities and complex data distributions of modern LLMs, weakening the direct applicability of the proofs.

---

> ### Author Rebuttal · Authors · 2026-03-31
>
> We thank the reviewer for your careful consideration. We greatly appreciate the positive comments and address your concerns below.
>
> **About Heuristic Guidelines**: We thank the reviewer for recognizing the comprehensiveness of our theoretical analysis. While our method does not introduce a new architectural variant, it goes beyond heuristic design. Specifically, our approach derives explicit, theory-grounded principles for hyperparameter selection and scaling from Kronecker SVD analysis, rather than relying on ad hoc choices.
>
> The Kronecker Adapters' design space is highly complex, involving multiple interdependent factors, including the dimensions and number of Kronecker components, scaling, and initialization.  Previous works[1-3] largely fix these design choices without justification. In contrast, our work comprehensively characterizes how these factors interact and translates this understanding into practical design rules.
>
> Our goal is therefore not to modify the underlying architecture, but to provide a principled framework for configuring it, which offers a theoretically grounded foundation for Kronecker Adapters. Empirically, these theoretically grounded choices lead to consistent performance improvements across diverse models and tasks.
>
> To further validate the robustness of our method, we conduct experiments on **massive-scale LLMs and vision transformers**. Please see the next section for the detailed results.
>
> **Results on Massive-scale LLMs and ViTs**： Following your suggestion, to evaluate the robustness of our proposed guidelines across different settings, we conduct experiments on **LLaMA-3-70B** for mathematical reasoning and **CLIP-ViT-B/16** for image classification. The detailed results are presented in Tables 1-4 in https://anonymous.4open.science/r/To-reviewer-Unc3-2384. It can be observed that our guidelines exhibit consistent effectiveness across different settings, demonstrating the robustness of our method. We are actively performing more studies on LLaMA-3-70B. While these experiments are progressing well, full results are pending due to time and computational resource constraints. We promise to share the appending results once the experiments complete during the discussion phase.
>
>
> **Simplified Settings**: We assume the linear model and isotropic sub-Gaussian data primarily for mathematical tractability. These assumptions are standard in seminal LoRA-related theoretical works[4-6]. Notably, even under this simplified setting, a rigorous theoretical analysis of KAs has been lacking due to the intrinsic complexity of the Kronecker structure. Our work fills this gap. Moreover, our theoretical findings are supported by empirical results, suggesting that the insights remain relevant beyond the simplified assumptions. To further address your concern, we conduct more experiments to verify the **theoretical principles in real-world settings**. We direct the reviewer's attention to the "Theoretical Principles in Real-World Settings" section in our response to Reviewer B4jt. The detailed results are presented in Tables 5-7 and Figures 1-3 in https://anonymous.4open.science/r/To-reviewer-Unc3-2384.
>
> **More Ablation Studies**: Following your suggestion, we conduct experiments to examine the performance gains of the stabilization scaling factor $\lambda$ versus the component design. The detailed results are presented in the following table. It can be observed that the performance gains of CDKA primarily stem from our component design, while the stabilization scaling factor further improves the performance under different component configurations. These results fully demonstrate the effectiveness of our method.
>
> | Method | GSM8k |
> |:---            |:---:|
> |KronA   |    $49.00_{\pm0.41}$  |
> |KronA + Stabilization Factor |    $49.43_{\pm0.37}$  |
> |KronA + Component Design |    $\underline{55.62}_{\pm0.39}$  |
> |KronA + Stabilization Factor + Component Design (CDKA)  | $\mathbf{56.71}_{\pm0.38}$      |
>
> [1] KronA: Parameter Efficient Tuning with Kronecker Adapter arXiv:2212.10650
>
> [2] MoKA: parameter efficiency fine-tuning via mixture of Kronecker product adaption COLING2025
>
> [3] MoKA: mixture of Kronecker adapters arXiv:2508.03527
>
> [4] LoRA+: Efficient Low Rank Adaptation of Large Models ICML2024
>
> [5] Riemannian Preconditioned LoRA for Fine-Tuning Foundation Models ICML2024
>
> [6] LoRA-One: One-Step Full Gradient Could Suffice for Fine-Tuning Large Language Models, Provably and Efficiently ICML2025

---

> > ### Author Rebuttal · Reviewer_Unc3 · 2026-04-03
> >
> > Thank you for your response. I will increase my score to weak accept.

---

> > > ### Author Response · Authors · 2026-04-04
> > >
> > > Thank you for your positive feedback and constructive suggestions. We have updated more experiments on LLaMA-3-70B, as we promised, in the same link. The results further prove the robustness of our proposed guidelines on massive-scale LLMs. We will incorporate the relevant discussions and experiments into the final version.

---

### Official Review · Reviewer_35Vh · 2026-03-12

**Soundness:** 2
**Presentation:** 3
**Significance:** 3
**Originality:** 2
**Overall Recommendation:** 4
**Confidence:** 3

**Summary:**

This paper studies the design of Kronecker adapters for parameter-efficient fine-tuning. The authors argue that the performance of Kronecker-based adapters is strongly influenced by the **component design**, i.e., the configuration of Kronecker factors \((r_1, r_2)\) and the number of components \(r\). The paper provides theoretical motivation based on Kronecker-SVD analysis and gradient alignment, suggesting several principles for component design. Based on these insights, the authors propose CDKA, a Kronecker adapter with improved component configuration and initialization. Experiments on GLUE and several NLG benchmarks show that CDKA achieves competitive or improved performance compared with existing PEFT methods while maintaining a very small number of trainable parameters.

**Compliance With Llm Reviewing Policy:**

Affirmed.

**Final Justification:**

I thank the authors for the rebuttal, I raised my score accordingly

**Key Questions For Authors:**

please see weakness part

**Limitations:**

yes

**Strengths And Weaknesses:**

### Strengths

- **Interesting perspective on Kronecker adapters.** The paper focuses on the often overlooked design choices of Kronecker components and provides useful insights into how component configurations affect performance.
- **Theoretical motivation.** The analysis based on Kronecker-SVD and gradient alignment offers a reasonable theoretical lens to study Kronecker adapter design.
- **Strong parameter efficiency.** CDKA achieves competitive performance with a very small number of trainable parameters compared with other PEFT approaches.
- **Clear empirical study.** The paper includes ablations examining the effects of component dimensions and number of components, helping to better understand design choices.

### Weaknesses

1. **Component study could be more comprehensive.**
   While the paper explores several component configurations, the explored design space appears limited. For example:
   - The degenerate case ($r_1=1$) is not studied. This configuration would reduce the Kronecker structure to grouped column scaling and could provide useful insight into the boundary between Kronecker adapters and simpler structured adapters.
   - Larger component dimensions (e.g., 32 or higher) are not explored, even though transformer weight matrices are typically very large.
   - The study only considers Kronecker products of two matrices. It would be interesting to explore higher-order Kronecker structures (e.g., $\(A \otimes B \otimes C\)$), which could provide additional flexibility in modeling structured updates.

2. **Limited model diversity in experiments.**
   Most experiments are conducted on LLaMA models (LLaMA-2 and LLaMA-3.1). It would strengthen the empirical evidence to evaluate the method on other model families (e.g., Qwen or other recent open models) to demonstrate robustness across architectures.

3. **Empirical results do not clearly support one theoretical claim.**
   The theoretical analysis suggests that increasing ($r_1=1$) may degrade performance. However, the empirical results in Table 3 appear relatively flat across different ($r_1=1$) values, and the best-performing configurations do not clearly follow this trend. The authors may clarify whether the observed differences are statistically meaningful or whether additional experiments are needed to better support this claim.

---

> ### Author Rebuttal · Authors · 2026-03-31
>
> We thank the reviewer for your careful consideration. We greatly appreciate the positive comments and address your concerns below.
>
> **More Component Studies:** Following your suggestion, we conduct further exploration on additional component configurations. Specifically, we investigate the cases of $r_1 = 1$, $r_1 \geq 32$ and $r_2 \geq 32$ as a supplement to Table 6 in our original manuscript. The detailed results are presented in the following table. It can be observed that these additional results further support the validity of our guidelines: a small $r_1 > 1$ yields improvements under the same parameter budget, while the degenerate case $r_1 = 1$ leads to degraded performance.
>
> Table 1. CDKA with different $r_1$ and $r_2$ under the same parameter budget.
> | $r_1, r_2, r$ | GSM8k |
> |:---                     |:---:|
> |1 1 8 (LoRA-One)            | $55.40._{\pm0.37}$ |
> |2 2 8                      |$\mathbf{56.71}_{\pm0.38}$|
> |16 16 8                     |$\underline{56.56}_{\pm0.64}$|
> |32 32 8                     |$55.32._{\pm0.40}$|
> |64 64 8                     |$55.70_{\pm0.50}$|
> |
> |1 16 1 | $54.99_{\pm0.62}$|
> |2 32 1 | $\mathbf{56.58}_{\pm0.53}$|
> |4 64 1 | $\underline{56.15}_{\pm0.09}$|
>
>
> **Higher-order Kronecker structures**: We thank the reviewer for the valuable suggestion of exploring higher-order Kronecker structures. However, our paper focuses on addressing the performance gap of the second-order Kronecker product defined in Eq. (2), which has recently attracted growing attention in the literature. To the best of our knowledge, this is the first comprehensive and fine-grained analysis of its component design and its impact on performance, supported by both theoretical insights and empirical results.
>
> While we greatly appreciate the reviewer for sharing this interesting idea, extending to higher-order Kronecker products, e.g., $A \otimes B \otimes C x$, requires reshaping the input vector $x$ into a tensor, which significantly increases both analytical and implementation complexity. We therefore consider this direction beyond the scope of our work.
>
> **Robustness of CDKA on More Models and Datasets**: Following your suggestion, we evaluate the performance of CDKA across different architectures and tasks, including **Qwen3-0.6B-Base** and **Qwen3-8B-Base** on mathematical reasoning and **CLIP-ViT-B/16** on image classification. We compare CDKA with LoRA-One, the state-of-the-art LoRA variant. The detailed results are presented in the following tables. It can be observed that **CDKA consistently outperforms LoRA-One across all settings**, demonstrating the robustness of our proposed method.
>
> Table 2. Qwen3-0.6B-Base and Qwen3-8B-Base on mathematical reasoning.
>
> | Method | Qwen3-0.6B-Base | Qwen3-8B-Base |
> |:---                     |:---:|:---:|
> |LoRA-One                  |$64.41_{\pm0.35}$|$86.09_{\pm0.34}$|
> |CDKA (Ours)               |$\mathbf{65.58}_{\pm 0.08}$|$\mathbf{86.70}_{\pm 0.04}$|
>
> Table 3. CLIP-ViT-B/16 on image classification task.
> | Method | Cars |DTD |EuroSAT |GTSRB |RESISC45 |SUN397 |SVHN| Average |
> |:---                     |:---:|:---:|:---:|:---:|:---:|:---:|:---:|:---:|
> |LoRA-One                   |82.68|77.77|**99.22**|98.42|95.48|75.34|97.34|89.46|
> |CDKA (Ours)                |**84.39**|**78.51**|99.00|**98.47**|**96.16**|**76.25**|**97.45**|**90.03**|
>
>
> **Additional Experiments on $r_1$**:  The reason the empirical results across different $r_1$ appear relatively flat is that although increasing $r_1$ hurts the alignment between CDKA and full fine-tuning, it does increase the attainable rank. Moreover, we choose a small $r_2=2$ in our experiments. As a result, the number of trainable parameters increases with $r_1 >2$, which in turn improves performance. To further address your concern, we conduct additional experiments with a larger $r_2 = 32$. The detailed results are presented in the following table. It can be observed that under this setting, **increasing $r_1$ consistently degrades the performance of CDKA**, even though it increases the attainable rank. We further conduct experiments on **Qwen3-0,6B-Base** and **CLIP-ViT-B/16**, with results provided in Table 1 in https://anonymous.4open.science/r/To-reviewer-35Vh-C10C. These empirical results further support our conclusion.
>
> Table 4. CDKA with different $r_1$ for fixed $r_2$ and $r$.
> | $r_1$ | GSM8k |
> |:---                     |:---:|
> |1                      |$\mathbf{55.29}_{\pm0.31}$|
> |4                      |$\underline{54.56}_{\pm0.54}$|
> |16                      |$51.78_{\pm1.15}$|
> |64                      |$47.49_{\pm1.37}$|

---

> > ### Author Rebuttal · Reviewer_35Vh · 2026-04-03
> >
> > thank you for the response

---

> > > ### Author Response · Authors · 2026-04-03
> > >
> > > Thank you for your thoughtful follow-up. We will incorporate the relevant discussions and experiments into the final version.

---

### Decision · Program_Chairs · 2026-04-30

**Decision:**

Accept (regular)

**Comment:**

This paper investigates the component design of Kronecker Adapters through the perspective of Kronecker Singular Value Decomposition. All reviewers reached a consensus that the work provides technically solid guidelines for hyperparameter configuration and a training stabilization strategy. During the rebuttal, the authors provided additional empirical results and ablation studies that successfully addressed the reviewers' concerns regarding the framework's generalizability and practical effectiveness. Therefore, based on the consistent positive assessments from the reviewers, the paper is recommended for acceptance.